# Simulated mixing in the UTLS by small-scale turbulence using multi-scale chemistry-climate model MECO(n)

Chun Hang Chau<sup>1</sup>, Peter Hoor<sup>1</sup>, and Holger Tost<sup>1</sup>

<sup>1</sup>Institute for Atmospheric Physics, Johannes Gutenberg University Mainz, Mainz, Germany

Correspondence: Chun Hang Chau (cchau@uni-mainz.de), Holger Tost (tosth@uni-mainz.de)

Abstract. The chemical composition of the upper troposphere/lower stratosphere (UTLS) plays an important role for the climate by affecting the radiation budget. Small-scale diabatic mixing like turbulence has a significant impact on the distribution of tracers which further affect the energy budget via their radiative impact. Current models usually have a higher vertical resolution near the surface and a coarser grid spacing in the free atmosphere, which is insufficent to resolve the occurrence of small-scale turbulence in the UTLS. In this work, we utilise enhanced vertical resolution (200 m in the UTLS) simulations focusing on mixing events in the Scandinavian region using the state-of-the-art multi-scale atmospheric chemistry model system MECO(n). These model simulations are able to represent different distinct turbulent mixing events in the UTLS and depict a significant impact of mixing on the tracer distribution in the UTLS. A novel diagnostic (delta tracer-tracer correlation) is introduced to determine the direction of the vertical mixing. The strength of the UTLS turbulent mixing depends on the particular situation, i.e., the vertical tracer gradient, and dynamical and thermodynamical forcing, i.e., vertical wind shear, deformation and static stability. This work provides evidence that high resolution simulations are able to represent significant turbulent mixing in the UTLS region, allowing for further research on the UTLS turbulent mixing and its implications for the climate system.

#### 1 Introduction

The upper troposphere and lower stratosphere (UTLS) is defined as the region around the tropopause which acts as a transition layer between the troposphere and stratosphere (Gettelman et al., 2011). The troposphere and stratosphere are fundamentally different in chemical composition and static stability, and they are separated by the tropopause, an immaterial surface acting as a vertical transport barrier. The dynamical tropopause (aka potential vorticity (PV) tropopause) is one of the commonly used definitions for the tropopause due to its conservation under isentropic conditions. The typical PV values for the dynamical tropopause can range from 1.6 PVU to 3.5 PVU, but 2 PVU is most commonly used (Stohl et al., 2003a). Since the PV-tropopause is a quasi-impermeable surface for adiabatic frictionless flow, i.e., on isentropes, stratosphere-troposphere exchange (STE) across the tropopause may require diabatic processes, e.g., like turbulent mixing by small scale turbulence (Holton et al., 1995).

The distribution of chemical constituents and the resulting changes in the UTLS chemistry are a consequence of the complex atmospheric processes on various spatial and temporal scales (Riese et al., 2012). Bi-directional STE is one of the crucial

processes affecting the chemistry of UTLS (Holton et al., 1995; Stohl et al., 2003b), especially in the extratropical transition layer (ExTL) in the extratropics (Gettelman et al., 2011).

The chemical composition of the UTLS plays an important role on the climate by affecting the radiation budget (?). Local changes in the tracer distribution in the UTLS will not only lead to local changes on the energy budget, but also affect the surface climate (Riese et al., 2012; Lacis et al., 1990; Randel et al., 2007). Previous studies showed that the surface temperature is highly sensitive to the changing chemical composition in the UTLS region (Forster and Shine, 1999, 2002). For example, changes in ozone distribution especially at the tropopause and lower stratosphere could have large impacts on the surface temperature (Forster and Shine, 1997). Besides of the radiation budget, STE also has impacts on other aspects, such as stratospheric ozone recovery (Butchart and Scaife, 2001) and the tropospheric ozone budget (Lelieveld and Dentener, 2000). Turbulent mixing is one of the processes of STE (Holton et al., 1995), especially in the region near the jet streams and tropopause folds (Shapiro, 1980). Clear air turbulence (CAT) is one of the major types of turbulence that occurs in the UTLS which could lead to rapid mixing of chemical species between stratosphere and troposphere (Esler and Polvani, 2004; Traub

tropopause folds (Shapiro, 1980). Clear air turbulence (CAT) is one of the major types of turbulence that occurs in the UTLS which could lead to rapid mixing of chemical species between stratosphere and troposphere (Esler and Polvani, 2004; Traub and Lelieveld, 2003). CAT refers to the turbulence in the free atmosphere that occurs in cloud-free regions or within stratiform clouds (Ellrod et al., 2003). It has a lifetime of an hour to a day, with a typical vertical dimension from 500 m to 1000 m (Overeem, 2002). Kelvin-Helmholtz instability (KHI) as a result of vertical shear of horizontal wind (Kunkel et al., 2019), forming a shear layer at the tropopause (Kaluza et al., 2021), is the major mechanism that leads to CAT formation (Watkins and Browning, 1973; Ellrod and Knapp, 1992). Consequently CAT occurs when the vertical wind shear is strong enough to overcome the stable layer's inhibition (Williams and Joshi, 2013).

CAT occurs most frequently in the UTLS, especially near the tropopause (Dutton and Panofsky, 1970; Wolff and Sharman, 2008) and along the jet streams (Keller, 1990; Traub and Lelieveld, 2003). This phenomenon shows the highest probability of occurence in boreal winter and is less frequent in boreal summer (Jaeger and Sprenger, 2007). An exceptional region is the eastern Mediterranean (Jaeger and Sprenger, 2007; Traub and Lelieveld, 2003) which is also known as a region with strong STE (Sprenger and Wernli, 2003).

Climate change is expected to increase the occurrence and intensity of CAT due to the strengthening of vertical wind shear (Williams, 2017), such that CAT is expected to have a large relative increase globally under the IPCC RCP 8.5 scenario especially in the mid-latitudes (Storer et al., 2017). Williams and Joshi (2013) results suggested that if the atmospheric CO2 is doubled compared to the pre-industrial time, the strength of CAT in the North Atlantic during winter will increase by 10-40% and the occurrence of CAT which is moderate or greater will increase 40-170%. Recent studies by Smith et al. (2023) and Hu et al. (2021) also show similar results over the Northern Atlantic and East Asia, respectively.

Considering the increasing trend of CAT, and the link between turbulent mixing and STE, and hence the radiation budget, it is crucial to investigate the relation between CAT and mixing of chemicals in the UTLS. However, previous studies mainly focus on the dynamical aspect of turbulence (Kaluza et al., 2021; Muñoz-Esparza et al., 2020), but not on tracers. The main objective of this study is not to analyse the representation and strength of the turbulence itself, but to systematically analyse the impact of turbulence on tracer mixing in the UTLS. For that purpose, a novel diagnostic, namely the delta tracer-tracer correlation is used within the multi-scale climate chemistry model MECO(n). Consequently, the main objective of the study is on the

resulting effects on the tracer distributions caused by turbulent mixing. Note to differentiate between the mixing itself, i.e., the "dynamical" mixing represented by e.g., the TKE, and the local effects on the tracer distributions provided by the mixing and the tracer gradient and further effects of the mixing, i.e., the downwind changes in the tracer distributions, originating from the mixing and subsequent processes, e.g., advection. Especially, the latter can further enhance vertical differences in tracer concentrations in case of modified vertical gradients of the respective tracers.

The paper is structured as outlined below. Section 2 introduces the applied model, describes the model configuration and the novel diagnostic delta tracer-tracer correlation. Crefsec:results presents the results and discusses the details of the mixing by passive tracer tests conducted in this study. Section 4 summarises the findings and draws conclusions.

#### 2 Model description and methodology

This section gives a brief introduction to MECO(n) (Mertens et al., 2016), the EMAC and COSMO set-up (including horizontal and vertical resolution, model domain and time step), the COSMO turbulence scheme, the explanation of the enhanced vertical grid for COSMO and the introduction of the novel diagnostic delta tracer-tracer correlation.

### 2.1 MECO(n) modelling system (v2.55.2)

MECO(n) represents the MESSy-fied (Modular Earth Submodel System, MESSy) European Centre Hamburg general circulation model (ECHAM) and Consortium for Small-scale Modeling (COSMO) models nested n times (Kerkweg and Jöckel, 2012a; Kerkweg et al., 2018), and is a state-of-the-art online coupled global/regional atmospheric chemistry model system based on the Modular Earth Submodel System (MESSy; Jöckel et al., 2005), which allows users to switch on or off physical and chemical processes through namelist interfaces. In MECO(n), the regional atmospheric model COSMO (Baldauf et al., 2011; Doms and Baldauf, 2018; Schättler et al., 2021) is nested online within the global general circulation model ECHAM5 (Roeckner et al., 2003); both COSMO and ECHAM5 are equipped with the MESSy infrastructure as individual COSMO/MESSy (Kerkweg and Jöckel, 2012b) and ECHAM/MESSy instances (EMAC; Jöckel et al., 2006; Jöckel et al., 2010). Besides the meteorological data, also the chemical composition and tracer information is exchanged between the individual instances. MECO(n) consequently allows an online coupling between different models so that the larger-scale (=parent, e.g., EMAC or COSMO/MESSy) instance can provide the initial and boundary conditions for the smaller-scale (= children, e.g., COSMO/MESSy) instances.

#### 2.2 MECO(n) model configuration

In this study, MECO(n) contains two smaller nests besides the global instance: EMAC is coupled with an intermediate COS-MO/MESSy instance (further denoted as CM40) and CM40 is further coupled with a target COSMO/MESSy instance (further denoted as CM10). EMAC is operated in T42L90MA (Giorgetta et al., 2006) resolution. It is a middle-atmosphere configuration that has 90 vertical layers up to 0.01 hPa (approximately 80 km in altitude) at T42 horizontal resolution (approximately  $2.8^{\circ} \times 2.8^{\circ}$  at the equator). The model time step is 360 s and it is initialized with the ERA-Interim reanalysis data (Dee et al.,

Figure 1. Model domain for MECO(n) with surface height: (a) an overview, (b) a close-up for CM40 and CM10.

2011). EMAC has been weakly nudged (Jeuken et al., 1996) towards the ERA-Interim reanalysis data up to 10 hPa. The CM40 domain covers most of Europe from Spain and Iceland in the west to parts of Russia in the east with a horizontal resolution of 0.4° and a model time step of 120 s. The initial and boundary data are provided by the EMAC instance. The CM10 model region focuses on the Scandinavian region with a horizontal resolution of 0.1° and a model time step of 40 s. The initial and boundary data are provided by the CM40 instance. Both CM40 and CM10 have 84 vertical layers with an enhanced resolution in the UTLS, details of the enhanced grid are discussed in Section 2.4.

#### 2.3 Vertical mixing in COSMO

The mixing in the COSMO model is divided into two parts: (1) small scale turbulent diffusion, and (2) organized moist convection. In this study, we focus on the impact of the small scale turbulent diffusion. In COSMO, the sub-grid scale turbulent diffusion is based on K-theory, the constitutive equation is as follows:

$$F_{\psi} = -K_{\psi} \cdot \nabla \psi$$

This equation relates the sub-grid scale turbulent flux of a scalar quantity  $F_{\psi}$  to the gradient of  $\psi$  and a diffusion coefficient  $K_{\psi}$ . The determination of the  $K_{\psi}$  depends on the chosen turbulent closure scheme. COSMO provides two different turbulent schemes. The default setup uses a 1-D diagnostic closure scheme by Muller (1981). In this scheme,  $K_{\psi}$  is determined by the Blackadar length scale (Blackadar, 1962), vertical wind shear, Brunt-Väisälä frequency and stability functions which are based on the flux-Richardson number. However, this scheme comes with several drawbacks including insufficient vertical mixing in stable stratification. COSMO also provides another newer turbulent scheme based on prognostic turbulent kinetic energy. The  $K_{\psi}$  in this prognostic TKE-based scheme is in the form of:

110 
$$K^H = q\lambda S^H$$

$$K^M = a\lambda S^M$$

where  $K^H$  and  $K^M$  are the turbulent diffusion coefficients for heat and momentum respectively. They are computed by the corresponding stability functions for scalars  $(S^H)$  and for momentum  $(S^M)$  which are determined by the flux-Richardson number, the turbulent length scale  $\lambda$  (which is assumed to be the Blackadar mixing length) and the turbulent velocity scale  $q = \sqrt{2\overline{e_t}}$  where  $\overline{e_t}$  is the turbulent kinetic energy (TKE). The latter scheme is used in this study. Details for the turbulent schemes can be found in section 3 (3.3.2 for the used scheme) of the COSMO model documentation by Doms et al. (2018).

**Figure 2.** The vertical level definition of D-40, D-50, EX-60 and EH-84 vertical grid, the shaded area represents the damping layer of the respective vertical grid.

#### 2.4 Enhanced vertical grid for COSMO instances

The default vertical grid for COSMO is either 40 or 50 levels that reach up to 22 km, with an 11 km damping layer starting at 11 km. Furthermore, these default vertical grids have a finer resolution near the surface and a coarser resolution in the free atmosphere, which makes the default setup too low and too coarse for resolving small scale turbulence or other processes in the UTLS. Previous studies also show that STE-related processes are sensitive to the model resolution (Miyazaki et al., 2010; Meloen et al., 2003; van Velthoven and Kelder, 1996). In MECO(n), the model TKE is sensitive to the vertical resolution and the mixing strength is sensitive to both horizontal and vertical resolution (details in supplement). Therefore, in this study, we

introduce an enhanced vertical grid focused on the UTLS which is applied to both CM40 and CM10. It is modified from an established extended vertical grid (Eckstein et al., 2015) with 60 levels (further denoted as EX-60) which reaches the lower stratosphere up to 33 km, with a 5 km damping layer starting at 27 km. Our enhanced setting has 84 levels and reaches also up to 33 km, with an identical 5 km damping layer starting at 27 km considering Eckstein et al. (2015) shows that the differences between a 5 km and 11 km damping layer is marginally small, and the analysis carried out in this study are far below the damping layer with more than 20 model levels, the potential reflection from the model top should be negligible. In order to reduce modifications of the boundary layer due to the change of vertical grid, we kept the levels below 8 km unchanged and only increase the resolution between 8-15 km to 200 m per level considering the typical size of CAT (Overeem, 2002). The level definition for the default 40 levels (further denoted as D-40), 50 levels (further denoted as D-50), EX-60 and the enhanced vertical grid (further denoted as EH-84) are shown in Figure 2. EH-84 is evaluated with ERA5 data, as well as comparisons with the tested EX-60 setup.

EH-84 is able to simulate the atmosphere reasonably. Although there is some discrepancy, the temperature pattern from ERA5 is generally well produced by the model as well as the relative humidity. There is a systematic cold bias in the CM10 output. However, the systematic cold bias that occurs in EH-84 is also found in EX-60 as well as the CM40 and EMAC output, indicating that the occurrence of the cold bias is not a result of the increased vertical resolution in the UTLS. There is a strong alignment of the main meteorological parameters between the EH-84 and EX-60 output, and the latter is well-evaluated against observations (Eckstein et al., 2015). Consequently, the model output from the enhanced vertical grid EH-84 can be seen as reliable and suitable for the needs of this study. Details for the evaluation of EH-84 can be found in the electronic supplement material. Furthermore, a sensitivity study was conducted in the supplement to show the necessity for higher vertical and horizontal resolution. The occurrence of high TKE values in EH-84 is more frequent than in EX60 (Figure S9), hence the mixing is more frequent. (Figure 8 and S11). CM10 (Figure 8) also shows more efficient mixing than CM40 (Figure S12) despite the similar TKE frequency.

# 2.5 Delta tracer-tracer correlation




In order to investigate the tracer mixing in the UTLS, we introduced a novel diagnostic, namely a delta tracer-tracer correlation, which is a similar concept to the tracer-tracer correlation, but makes use of the model capabilities. While the tracer-tracer correlation can be compared to the real world, the delta tracer-tracer correlation is a correlation between the differences of the tracers from model experiments. Instead of showing the mixing as an accumulation affected by other processes, it shows the impact of a single process (and potential subsequent advection differences). It requires 2 pairs of tracers (one pair of stratospheric and one pair of tropospheric). The difference of each pair, is a particular process being deactivated on one of the tracers in the model to investigate the impact of it. In our study, it is the turbulent vertical diffusion (vdiff). The detailed released tracers are described in Section 3.2. The delta tracer-tracer correlation can also be used to determine the direction of vertical mixing. Several distributions are expected for different scenarios: (1) Concentrated distribution at the center [0,0] if no vertical mixing takes place at all; (2) Diagonal distribution for bi-directional mixing, where both tracers change at a similar rate, causing the data point spread along the diagonal. The bi-directional mixing could be either balanced or imbalanced,

meaning an even (case 1, spread equally from the center [0,0]) or uneven (case 2, spread unequally from the center [0,0]) spread along the diagonal. Balanced bi-directional mixing indicates a similar amount of stratospheric tracers being exchanged with the tropospheric tracers, while imbalanced bi-directional mixing indicates a different amount of stratospheric tracers being exchanged with the tropospheric tracers. It could be attributed to different situations, details are discussed in the following cases. The upper left section of the diagram indicates the downward mixing of stratospheric tracers into the troposphere since at the same grid, there are increasing stratospheric tracers and decreasing tropospheric tracers. And the lower right quadrant indicates the opposite, with decreasing stratospheric tracers and increasing tropospheric tracers i.e. upward mixing of the tropospheric tracers. Scatter further away from the center indicates irreversible mixing, as the composition of the air masses is substantially modified, and the tracer is mixed irreversibly into the grid, i.e., instantaneously horizontally mixed. Additionally, this scatter is caused by initial differences from the mixing which are then amplified by (mostly horizontal) advection into regions where the vertical gradient of the tracers are different. Those different gradients can originate both from the tracer mixing event itself further upstream or from specific meteorological conditions, e.g., tropopause folds with strong gradients. Scatter away from the diagonal (case 3) indicates that the mixing occurs in a region with a different tracer gradient, a non-local effect introduced by other processes like horizontal advection acted on the mixed tracer. The scatter away from the diagonal gives an indicator that the mixing is non-local but the strength of mixing itself is still solely contributed by the turbulent mixing.

#### 3 Results and Discussion





#### 175 3.1 Turbulence in the UTLS

Considering that turbulence in EMAC is dampened in the free atmosphere due to its hydrostatic characteristic and the formulation of the turbulence scheme (designed for the boundary layer only), this section analyses how well the COSMO instances are able to represent turbulence and associated mixing. Therefore, the model turbulence kinetic energy (TKE) is compared with a calculated turbulence index using the grid-scale wind data from COSMO, i.e., the turbulence diagnostic TI1 from Ellrod and Knapp (1992), which includes a vertical wind shear term and a deformation (stretching and shearing) term to examine whether the highly parametrized subgrid scale turbulence scheme is consistent with the grid-scale wind. The calculated TI1 is divided into 5 categories (i.e. null, light, moderate, severe and extreme) according to the thresholds set by Sharman et al. (2006). The features of the turbulence including the distribution and relative strength are reproduced by the COSMO instance as can be seen in Figure 3 and Figure 4, which shows the calculated TI1 (Figure 3) and model TKE (Figure 4) on the selected vertical levels, respectively. The TI1 generally agrees with TKE in terms of distribution and relative strength. The discrepancy between them might be caused by the neglected mechanisms of the Ellrod index or other sub-grid scale processes that could potentially lead to the formation of turbulence in the UTLS, e.g., sub-grid scale gravity waves. The high TKE and Ellrod index shown in Figure 3 and Figure 4 is caused by the jet stream as shown in Figure S18. The increased shear could be attributed to the higher vertical and horizontal resolution, as shown in Figure S19. CM10 shows more fine structure and hence more shear. It is important to note that the Ellrod index does not fully represent the turbulence in the atmosphere since it does not account for all producing mechanisms. For example TI1 might neglect the shear related to anticyclonic flow (Ellrod and Knox, 2010).

It is also important to note that the comparison between the TI1 and TKE is not sufficient enough considering both of them are calculated from the COSMO wind field; comparison with observation should be conducted when there is available data. In addition, the strength between TI1 and TKE is not directly comparable since the TI1 threshold was set according to the verbal report of pilots and is subjective to the pilot's feelings and there is no similar threshold available for TKE. However, the results at least show consistency in the distribution on different levels. To conclude, the model grid scale wind field is consistent with the model turbulence scheme and can detect the occurrence of turbulence in the model.



**Figure 3.** Calculated Turbulence Index (TI1) at 2016-02-07 20:00; null = Grey, green = light, yellow = moderate, orange = severe, red = extreme.

We also compare the model results with the last flight in the GW-LCYCLE II campaign (Witschas et al., 2023) on the 1st of February in northern Scandinavia. We derive a measure of turbulence from the high frequency measured  $N_2O$  (Lachnitt et al., 2023) and link it with the model TKE. We computed a 31-point running standard deviation normalized with the variability of the window for  $N_2O$ . The running standard deviation shows the  $N_2O$  fluctuation from the background in a short period of time while the normalization eliminates the effect of a tracer gradient due to the changing flight altitude or large-scale exchange of air masses. Figure 5 shows the model TKE at the flight time with the normalized running standard deviation of the measured

Figure 4. Model Turbulence kinetic energy (TKE) at 2016-02-07 20:00.

N<sub>2</sub>O. It shows that the derived turbulence signal often coincides well with the simulated TKE (Figure 5a), the stronger signals (higher percentiles, Figure 5b) coincide with the higher model TKE as well. This indicates that there is a reasonable degree of consistency between the derived turbulence signal from the measured N<sub>2</sub>O with the simulated turbulence.

## 3.2 Passive tracer test


In order to investigate the ability of mixing by turbulence in MECO(n), a series of passive tracer tests is performed by initializing several pairs of passive tracers in the simulation via the MESSy submodel PTRAC (Jöckel et al., 2008). The PTRAC submodel allows users to define the physical and chemical properties of specific tracers. In this study, we define a total of 4 pairs of artificial passive tracers with different distributions and slightly different physical properties. For the same pair of tracers, the only difference is whether the physical process of vertical diffusion (vdiff) is turned on or off. The vertical diffusion was switched off at the very beginning. An O<sub>3</sub>-like tracer with a relatively steep linear gradient and a N<sub>2</sub>O-like tracer with a relatively gentle gradient are initialised to investigate the effect of the tracer gradient on the strength of mixing under a relatively

Figure 5. Cross section of average Model Turbulence kinetic energy (TKE) during the flight time. The black line represents the PV-tropopause, grey line represents the flight track, scatters represent the (a) 99 percentile and (b) 99.9 percentile of the normalized running standard deviation, i.e. the regions where the  $N_2O$  shows high normalised variability, potentially due to turbulence.

**Table 1.** Summary of the released passive tracers

| Pair no. | Tracer                        | vdiff(on/off) | Mixing ratio [mol/mol] | Stratospheric/Tropospheric |
|----------|-------------------------------|---------------|------------------------|----------------------------|
| 1        | O <sub>3</sub> -like          | on            | 2.4e-08 to 4.0e-06     | Stratospheric              |
| 1        | O <sub>3</sub> -like          | off           | 2.4e-08 to 4.0e-06     | Stratospheric              |
| 2        | N <sub>2</sub> O-like         | on            | 3.2e-07 to 6.4e-08     | Tropospheric               |
| 2        | N <sub>2</sub> O-like         | off           | 3.2e-07 to 6.4e-08     | Tropospheric               |
| 3        | Inverted O <sub>3</sub> -like | on            | 4.0e-06 to 2.4e-08     | Tropospheric               |
| 3        | Inverted O <sub>3</sub> -like | off           | 4.0e-06 to 2.4e-08     | Tropospheric               |
| 4        | Inverted N2O-like             | on            | 6.4e-08 to 3.2e-07     | Stratospheric              |
| 4        | Inverted N2O-like             | off           | 6.4e-08 to 3.2e-07     | Stratospheric              |

tively realistic scenario. The tracers change linearly at the transition layer near the tropopause between approximately 300 hPa to 100 hPa. The initial condition of the tracers is shown in figure S20 as a vertical profile. In order to investigate the direction of mixing, inverted versions of both tracers are also released in order to have stratospheric and tropospheric tracers with a similar gradient at the same time. A summary of the tracers is shown in Table 1.

**Figure 6.** Case 1: Geopotential height (gpm) at 200 hPa from CM40 at 2016-02-05 18:00. Green line indicates the location of the selected cross section of case 1.

**Figure 7.** Case 1: (a) CM10 horizontal wind speed, the black contour line show the O3-like tracer and (b) gradient Richardson number (Ri), the black line indicates the PV tropopause.

**Figure 8.** Cross section of distribution (a) Inverted O3-like tracers (mol/mol), (b) O3-like tracers (mol/mol); and (c) difference (vdiff on off) of the Inverted O3-like tracers (mol/mol), (d) difference of the O3-like tracers (mol/mol) at 2016-02-05 18:00. The black line indicates the PV-tropopause.

#### 3.3 Results: Case studies



## 3.3.1 Case 1: turbulence induced balanced bi-directional mixing in stable region

Case 1 (Figure 6) located within a typical high level ridge trough system over Europe, at the transition region between the anticyclonic ridge and the cyclonic trough, with the potential of strong wind shear and convergence. It is also associated with the jet stream, regions with relatively low Ri are found at the vicinity of the tropopause and the core of the jet stream (Figure 7). Figure 8 shows the cross section of the distribution (top) and differences (bottom; vdiff on - vdiff off) for the O3-like (right) and inverted O3-like (left) tracers. The results show that vertical turbulent diffusion has a significant impact on the tracers. For the tropopheric inverted O3-like tracers, a higher mixing ratio above the tropopause and a lower mixing ratio below the tropopause is simulated when vertical turbulent diffusion is present. This indicates that the tracers were transported across the

tropopause by turbulent mixing from the troposphere to the stratosphere. The stratospheric O3-like tracer shows analogous

behavior but in an inverse manner, in which the turbulent mixing shifts the tracers from the stratosphere into the troposphere. By comparing the differences with the background mixing ratio, vertical mixing could lead to almost 10% of differences near the tropopause. Similar mixing behavior is also noticeable for the N2O-like and inverted N2O-like tracers but in a weaker form (approximately 5%) due to its relatively gentle gradient (Figure S8). Note, that the tracer differences are strongest, exactly in the region with the lowest gradient Richardson number (cf. Fig. 7b and Fig .8c,d), indicating turbulence as the origin of the high spatial correlation.

Figure 9. tracer-tracer correlation for (a) O3-like/N2O-like tracers with vdiff (mol/mol); (b) O3-like/N2O-like tracers without vdiff (mol/mol) at 2016-02-05 18:00.

**Figure 10.** Case 1: Delta tracer-tracer correlation for determining the direction of vertical mixing of stratospheric O3-like/Inverted tropospheric O3-like tracers (mol/mol) color-coded with (a) Ellrod Index ( $s^{-2}$ ), (b) vertical wind shear ( $s^{-1}$ ), (c) deformation ( $s^{-1}$ ), (d) Brunt–Väisälä frequency ( $s^{-2}$ ) and (e) turbulence kinetic energy ( $m^2 s^{-2}$ ) at 2016-02-05 18:00.



Figure 9 shows the tracer-tracer correlation for different pairs of passive tracers at the same time and location as the cross section of Figure 8. Figure 9a and b show a tracer-tracer correlation between the O3-like stratospheric tracer and N2O-like tropospheric tracer with and without vertical diffusion respectively. Considering the passive tracers were released with a linear gradient, the tracer-tracer correlation shows a linear distribution as well, unlike the other classic tracer-tracer correlation which normally has an exponential relationship. Perfect correlation with diagonal distribution is expected if vertical diffusion does

not play any role in transporting the tracer. Considering the magnitudes of the mixing ratio in both tracers, the difference is hard to distinguish for a single mixing event of Figure 8 in the tracer-tracer correlation. Therefore, the tracer-tracer correlation of the same tracer with and without vertical diffusion was performed as well. Figure 9c and d show the correlation with and without vertical diffusion for the stratospheric O3-like and tropospheric N2O-like tracer respectively. Both tracers show some dispersion from the diagonal, indicating that vertical diffusion is affecting the tracers, leading to a deviation from perfect correlation. In addition, a delta tracer-tracer correlation (see Section 2.5 for an explanation of this diagram) is performed for O3-like and N2O-like tracers.

Figure 10 shows the color-coded delta tracer-tracer correlation for the stratospheric O3-like/Inverted tropospheric O3-like tracers of the mixing event at 2016-02-05 18:00:00 (Figure 8). It is conducted using every grid point between 100 hPa to 350 hPa at the indicated location on Figure 6. It is a bi-directional mixing event associated with turbulence, which shows a diagonal distribution, indicating that at a specific location, the change of stratospheric tracer is similar to the change of tropospheric tracer. The symmetric distribution indicates that the mixing is balanced in strength in both directions. The strong downward mixing of the stratospheric air is caused by vertical wind shear or/and deformation: the region with strong downward mixing is concurrently the region with extreme turbulence according to the Ellrod index, as well as the strong vertical wind shear (VWS), deformation and relatively high TKE values. Considering that the vertical wind shear and deformation are the key mechanisms for turbulence formation, and vertical wind shear is related to the calculation of TKE, it is reasonable that they show similar behavior. For static stability, the Brunt-Väisälä frequency  $(N^2)$  shows no distinct behavior, with most of the region reaching the typical stratospheric value. These characteristics of the mixing event are consistent with the findings by Kaluza et al. (2021), where strong vertical wind shear is able to be maintained under stable conditions. The strong upward mixing of the tropospheric air cannot be easily attributed to the vertical wind shear or deformation. Although light turbulence occurs in the strong upward mixing regions, the same strength of mixing as the downward flow cannot be explained. According to the constitutive equation of vertical diffusion in Section 2.3, the turbulent flux of tracers is calculated by the diffusion coefficient and the gradient of the tracer. Besides the diffusion coefficient, which is determined by the dynamics and thermodynamics of the atmosphere, the tracer gradient also plays a role on the mixing strength, such that mixing in a homogeneous atmosphere will have no effects on the tracers no matter how strong the mixing coefficient would be. In order to investigate the impact of the tracer gradient, the mixing is normalized by the tracer gradient to remove its impact. Figure 11 shows the same delta tracer-tracer plot but colorcoded with absolute value of the difference of the stratospheric O3-like tracer (left, |dO3ST|) and absolute value normalized with the tracer gradient (right, IdO3STI / Igradientl). The downward mixing attributed to the dynamical forcing remains strong after normalization while the upward mixing with much weaker dynamical forcing became weaker compared to the downward flow after normalization, showing that the upward flow could be attributed to the tracer gradient.

## 3.3.2 Case 2: Imbalanced bi-directional Mixing





Case 2 (Figure 12) again located within a typical high level ridge trough system over Europe, but instead of located within the transition region as case 1, it located closer to the ridge axis. It is also associated with the jet stream, region with relatively low Ri are found at the vicinity of the tropopause and jet stream (Figure 13).

Figure 11. Delta tracer-tracer correlation color-coded with IdO3STI (mol/mol; left) and IdO3STI/Igradientl (right)

**Figure 12.** Case 2: Geopotential height (gpm) at 200 hPa from CM40 at 2016-02-05 05:00. Blue line indicates the location of the selected cross section of case 2.

Figure 13. Case 2: (a) CM10 horizontal wind speed, the black contour line show the O3-like tracer and (b) gradient Richardson number (Ri), the black line indicates the PV tropopause.

Figure 14. Case 2: Delta tracer-tracer correlation of stratospheric O3-like/Inverted tropospheric O3-like tracers (mol/mol) color-coded with (a) Ellrod Index  $(s^{-2})$ , (b) vertical wind shear  $(s^{-1})$ , (c) deformation  $(s^{-1})$ , (d) Brunt–Väisälä frequency  $(s^{-2})$  and (e) turbulence kinetic energy  $(m^2 s^{-2})$  at 2016-02-05 05:00.

Figure 14 shows the similar plot as Figure 10, however, this time for an imbalanced bi-directional mixing event at 2016-02-05 05:00:00. The graph also shows a diagonal distribution, but asymmetrically. The lower right have a significantly shorter range than the upper left (unlike Case 1, which has a similar range on both ends). This indicates that in this specific profile, the changes of stratospheric air are different from the tropospheric air. This is a consequence of asymmetric stability and flow conditions, i.e., the stable layering of the stratosphere prevents deeper mixing into the stratosphere, whereas the lower static stability in the troposphere allows for deeper penetration of stratospheric tracers into the troposphere (where in Figure 14d, the lower right with high  $N^2$  have a shorter range than the upper left with low  $N^2$ , while Figure 10d of case 1 have a similar  $N^2$  on both ends).

The mixing strength in this case is relatively weak compared to the other cases. The stronger downward mixing of stratospheric air could again be attributed to the relatively strong vertical wind shear and deformation, most of the region with downward mixing is at least experiencing light to moderate turbulence. The low static stability also plays a role for the stronger downward mixing. The region with weaker upward mixing exhibits noticeably weaker vertical wind shear and deformation compared to the region with downward mixing. The atmosphere is also much more stably stratified than the region with strong downward mixing (the  $N^2$  is distinctly higher in this region). The upward mixing tropospheric air is therefore weaker because the weak dynamical instability is suppressed by the strong static stability.

**Figure 15.** Case 3: Geopotential height (gpm) at 200 hPa from CM40 at 2016-02-03 22:00. Purple line indicates the location of the selected cross section of case 3.

**Figure 16.** Case 3: (a) CM10 horizontal wind speed, the black contour line show the O3-like tracer and (b) gradient Richardson number (Ri), the black line indicates the PV tropopause.

Figure 17. Case 3: Delta tracer-tracer correlation of stratospheric O3-like/Inverted tropospheric O3-like tracers (mol/mol) color-coded with (a) Ellrod Index  $(s^{-2})$ , (b) vertical wind shear  $(s^{-1})$ , (c) deformation  $(s^{-1})$ , (d) Brunt–Väisälä frequency  $(s^{-2})$  and (e) turbulence kinetic energy  $(m^2 \, s^{-2})$  at 2016-02-03 22:00.

## 3.3.3 Case 3: Mixing associated with strong vertical gradient

Case 3 (Figure 15) is located at the outflow region of the high level trough ridge system, with potential of strong divergence. It is also the only case that the jet stream was shifted outside of the CM10 model domain, causing it relatively stable compare to the other two cases (Figure 16).

Figure 17 shows another mixing event associated with strong vertical gradient at 2016-02-03 22:00:00. The mixing again shows a diagonal and symmetric distribution, but with more scatter from the diagonal. This means, that the mixing does not only lead to equal changes in the tracer distributions, but more to entries of tropospheric tracers into regions of typically stratospherically dominated regimes. The scatter away from the diagonal unlike the other two cases, where the modeled TKE is better correlated with the mixing (not shown), is due to the advection, the mixing shown in Figure S17 and S22 located at the downwind region of the high TKE region (Figure S21). In the earlier time, the mixing region (Figure S22, left panel) is more co-located with the high TKE region (Figure S21, left panel). After several hours, the mixing region (Figure S22, right panel) propagates to the downwind region while the high TKE region (Figure S21, right panel) remains at the same location. The strong horizontal advection in the region of strong horizontal gradients changes the background ratios in addition to the vertical mixing and thus introduces additional mixing during each time step compared to the other cases. The wider the scatter is, the more, e.g., tropospheric tracer depletion is found at similar stratospheric tracer values.

However, in contrast to case 1, the dynamical and thermodynamical forcing do not play a key role in this case. The Ellrod index shows nearly no turbulence at all, neither vertical wind shear nor deformation shows any distinct behavior as in case 1. The static stability does not reach very high values in the stratosphere, such that the mixing is almost equally balanced.

#### 3.3.4 Case inter-comparison







In order to examine whether the tracer gradient is responsible for the strength of the mixing events, the mixing is again normalized by the tracer gradient. Figure 18 and Figure 19 show the frequency distribution for all 3 cases before (lmixingl) and after (lmixingl / lgradientl) normalization. Cases 1 and 3 have similar strength on mixing while case 2 is significantly weaker. Moreover, Cases 1 and 2 have similar distributions on dynamical forcing whereas case 3 forcing is notably weaker. After normalization, the mixing of Case 3 becomes much weaker considering the dynamical forcing does not play much role, proving that the vertical tracer gradient is responsible for the mixing in this case. Case 1 also becomes relatively weaker as expected since the downward mixing is attributed to the tracer gradient. The weakest case 2 turns out to be the strongest case without the impact of the tracer gradient.

To conclude, vertical turbulent mixing by CAT in the model simulations leads to an enhanced and significant tracer mixing in the UTLS region. The strength and direction of the mixing depends on the particular situation, whether the tracer gradient or the dynamic and thermodynamics of the atmosphere play a role. The tracer gradient plays the most important role since mixing will be meaningless if there is no tracer gradient. This confirms the findings of Kaluza et al. (2021) and Kunkel et al. (2019) that strong dynamical forcing like vertical wind shear could lead to mixing even in the stable atmosphere with a typical stratospheric  $N^2$  value.

**Figure 18.** Frequency distribution of ldO3STl (mol/mol) of O3-like tracers, vertical wind shear, deformation, TKE and N2 for case 1 (top), case 2 (middle), case 3 (bottom)

**Figure 19.** Frequency distribution of ldO3STl/lgradientl of O3-like tracers, vertical wind shear, deformation, TKE and N2 for case 1 (top), case 2 (middle), case 3 (bottom)

#### 4 Conclusions







This study presents model simulations for vertical tracer mixing in the UTLS region. The simulation configuration with an enhanced vertical resolution in the UTLS allows a more detailed analysis of turbulent mixing in this region and provides a suitable tool in the future understanding and quantification of the bi-directional cross-tropopause transport with implications on the Earth's radiation budget. In this work, a new enhanced vertical resolution model setup (~ 200m vertical resolution in the UTLS) for the regional model COSMO, which is nested within the multi-scale climate chemistry model MECO(n), is presented. It performs similar to established configurations and the ERA5 reanalysis with respect to large scale temperature and humidity fields in the UTLS, but allows a better representation and analysis of turbulent mixing events in this region. Within the relatively short simulation period, the simulations are able to capture several distinct turbulent mixing events in the UTLS with different characteristics including balanced and imbalanced bi-direction mixing induced by turbulence and strong vertical tracer gradient. The simulated turbulent kinetic energy (TKE) is spatially and temporally well matched with the (post-simulation) diagnosed Ellrod Index, showing the model is able to generate turbulence in the UTLS in agreement with the gridscale wind field data from the model output. The derived turbulence signal from N2O also shows a reasonable consistency with the simulated turbulence. However, further comparison with observation is needed considering both the modeled TKE and diagnosed Ellrod Index are calculated from the COSMO wind field. This model turbulence is able to significantly mix trace species vertically, as analysed from the changes in the vertical distribution of passive tracers. However, individual mixing events depend on the particular weather situation, for example, the vicinity of a jet stream which located near the tropopause experiencing the strongest mixing due to the high vertical wind shear and tracer gradient (case 1). However, it remains challenging to determine how well the model mixing strength is compared to the real world. Further analysis with measurement data is needed when a more comprehensive measurement dataset is available.

The diagnostic of a delta tracer-tracer correlation is used for the analysis of model simulations, in which the correlation of tracer differences between simulations with and without a representation of the turbulent mixing in the UTLS of stratospheric and tropospheric tracers are compared against each other. Both, the vertical tracer gradient and the dynamic and thermodynamic forcing, i.e., the stability and stratification, play important roles in the strength of vertical species exchange, especially when the vertical wind shear is strong enough to overcome the stable atmosphere. Depending on the individual situation, either the dynamical forcing or pre-existing tracer gradients (or both) can be the dominant drivers for the exchange events. The favorable combination of both factors can lead to an efficient mixing event, maximising tracer exchange fluxes. These events can be irreversible, i.e., the exchange of tracers happens along the diagonal of a delta tracer-tracer correlation, leading to a disturbance of typical stratospheric or tropospheric chemical compositions in the respective parts of the atmosphere with implications for climate, e.g., via the radiative impact of exchanged species.

Code availability. The model code of the MECO(n) system can be obtained by becoming a member of the EMAC consortium as described on the corresponding webpage https://messy-interface.org/.

*Author contributions*. CHC and HT conceptualised the study with contributions from PH. CHC performed the simulations and analysed the model results. The results were interpreted by CHC, HT and PH. CHC wrote the article with significant input from HT and PH.

Competing interests. The contact author has declared that none of the authors has any competing interests

Acknowledgements. This work has been funded by the Deutsche Forschungsgemeinschaft (DFG, German Research Foundation) – TRR 360 301 – Project-ID 428312742 (project B01). The simulations were conducted using the supercomputer MOGON II of Johannes Gutenberg University Mainz (https://hpc.uni-mainz.de/, last access: 15 November 2024).

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
