# Peer review of "Simulated mixing in the UTLS by small-scale turbulence using multi-scale chemistry-climate model MECO(n)"

_EGUsphere, 2024_

## Referee Comment (RC2)

**Simulated mixing in the UTLS by small-scale turbulence using multi-scale chemistry-climate model MECO(n)**

**by Chau et al.**

**Summary:**

Chau et al. present in their study the mixing in the upper troposphere / lower stratosphere (UTLS) region, based on simulations with the multi-scale chemistry-climate model MECO(n). The study is interesting and very well written, contributing with novel results and insights in the turbulent processes leading to the mixing in the UTLS. The statements in the paper are well supported by the figures, which are of high quality. I have no doubt that the study will be of high interest to the community dealing with stratosphere troposphere exchange (STE) and the chemical and dynamical (stability) 'structure' of the UTLS region and the tropopause.

There are only few (major) comments that I invite the authors to consider. In particular, these are:

1) The passive-tracer test in Section 3.2 relies only on one turbulence indicator (Ellrod and Knapp's TI1), whereas there are many other indicators available (e.g., Richardson number as an indicator for Kelvin-Helmholtz instability, KHI). Hence, the 'validation' against this turbulence indicator must be considered with some caution. Furthermore, the turbulence indicators (TI1, TI2, Richardson number,...) are also not the truth, but may fail in recognizing turbulence and/or indicate turbulence where actually there is none. I suggest that the authors more carefully discuss the limitations (and potential) of the turbulence indicators in the passive-tracer test.

2) The three case studies, particularly the discussing of the delta tracer-tracer correlations are very interesting and inspiring! There are two aspects that might be improved, however. First, Subsection 3.3.1 comes with a short introduction on this type of diagram, which is nice. I think that the reader can even be more clearly be introduced on how to read these diagrams. A few additional sentences might be helpful. For instance, why is an L-shape indicative for asymmetric mixing, whereas a diagonal one is clearly pointing to symmetric mixing; where in these diagrams do we see mixing from stratosphere to the troposphere and vice versa. See also the specific comments below. Second, the three cases in Section 3.3 discuss in detail the mixing and tracer-tracer correlations, which is the key topic of the study. Still, it would also be interesting (at least to me) to learn somewhat more about the dynamical and thermodynamical 'setting' of the cases. For example, how does the wind field look like? How is the vertical stratification? Or, why do we see in Figure 5 mixing at a very distrinct location (blue/red dipole) but not at neighboring locations. I fully understand that these aspects are not the key topic of the paper, still some mesoscale context would be interesting.

3) A minor point on structure: I wonder whether Section 3.3.4 is actually needed, or whether the aspect of mixing normalization (to determine the importance of gradient vs. dynamically forced mixing) would better integrated in the three case studies. In short, the discussion of this effect is interesting, but maybe not necessarily in an onw subsection. In line with this, I wonder whether Figures 11 and 12 are needed in the main text, or could become part of some supplementary material? I have not a very strong opinion on that, but the authors might consider it. Furthermore, the statements in Section 4 (Conclusions) are rather general and vague. Some examples: L245 ("with different characteristics" → which?); L249 ("However, individual mixing events depend on the particular weather situation" → what

weather situation? Which ones have been studied in this paper?); L55 ("Both, the vertical tracer gradient and the dynamic and thermodynamic forcing, i.e. the stability and stratification, play important roles in the strength of vertical species exchange" → Maybe, that what we already would have expected! So, can you make a more nuanced statement about the relative importance?).

**Specific Comments:**

- L17: aka PV tropopause; introduce 'potential vorticiy' before referring to it as PV

- L19: 'PV-tropopause is conserved' → I understand what the authors mean: that PV is conserved under adiabatic, frictionless flow conditions, and that thus the PV-tropopause behaves as a material surface under the these assumptions. The authors might rephrase, as 'tropopause is conserved' sounds a little strange.

- L27: affecting → affects

- Figure 1: Are really both panels needed?

- L88: COSMO(tbc) → What does 'tbc' mean?

- L95-98: Here, first the default setup of the 1D closure is described, whereas in the following text the actually used TKE-based scheme is described. Is it really necessary to describe the default scheme when it is not used? I see the authors' point to highlight in which way the default setup has been improved, but still wonder whether it is needed…

- L103-106: As before, do we need such a detailed description of the default setting? Or would be sufficient to just highlight in the text the improved setup?

- L116: Very minor detail: I think it should be ERA5 instead of ERA-5; and possibly start a new paragraph after 'EX-60 setup'?

- Section 3.1 (as discussed above): To which degree does this analysis depend on the specific section of the TI1 turbulence indicator? To which degree can the TI1 threshold to be assumed independent from the model resolution? Would you expect different results if TI2 and/or Richardson number is used instead of TI1? No detailed analysis is needed in my opinion, but a more critical discussion of the comparison betweem TI1 and the model TKE.

- Figure 5 (and corresponding text): I like the discussion on the tracer and inverted-tracer mixing, but wonder as a dynamic meteorologist why the mixingf only occur where it does and not at the other locations. Some meteorological background/context for the mixing events would be interesting, possibly supported by figures in the supplementary material?

- Figure 6 (panels c an d): Whereas panels a) and b) show straight diagonals, panels c) and d) exhibit some off-diagonal deviations. I wonder whether I correctly understand these. As an example, in panel c) in the lower-left quarter O3 with vdiff corresponds to somewhat below-diagonal O3 without vdiff. Does this mean that mixing (with vdiff) lowers the O3 values?

- Introductory paragraph to Section 3.3.1: Possibly, introduce the reader even more carefully how to read the delta tracer-tracer plots (as explained above). Why does an L-shale indicate asymmetric

mixing? Where in the plots do we see mixing from the stratosphere into the troposphere, where in the opposite direction? Somewhat refelcting on it, it becomes obvious? But why not helping the reader not familiar with tracer-tracer plots with 1-2 explaining sentences to get it at first reading?!

- L181: Minor detail; maybe start new paragraph

- L 185: 'by dynamic instability' → The term 'dynamic instability' is somewhat vague/unclear? I suggest that the authors introduce at some point in the manuscript very clearly what they mean with 'dynamic instability', and also set into contrast to other instabilities? Hence, if it is not a dynamic instability, what is it then? I assume that 'dynamic' refers essentially to the ones where wind (horizontal and vertical shear/deformation) is important, whereas a 'static instability' refers to one where a vertical profile becomes (thermo-dynamically) becomes unstable (e.g., if squared Brunt-Väisällä frequency becomes negative)? In essence, I suggest to carefully introduce these terms…

- L199: I get the basic idea, but what does |mixing| and |mixing| / |gradient| exactly stand for?

- L206-207: "This is a consequence of asymmetric strength and extension of the dynamical forcing, i.e., the stable layering of the stratosphere prevents deeper mixing into the stratosphere, whereas the lower static stability in the troposphere allows for deeper penetration of stratospheric tracers into the troposphere" →  I am not sure whether I fully understand this argument. First, what exactly is 'dynamical forcing' referring to? The discussion in the 'i.e.' sentence is more about stability than on dynamics. Second, the asymmetry argument would also be true in case 1?! Hence, also there the stably stratified stratosphere inhibits a deeper mixing than the more weakly stratified troposphere? Maybe, the solution to this is in the comparison of Figure 9d) with Figure 10d), which shows squared N and clearly differs between the two? In short, some further explanations might help (1-2 sentences).

- L218-221: Please explain in greater detail how the scatter from the diagonal relates to irreversible mixing. I think I kind of get the basic idea, but some further details could be helpful.

- Section 3.3.4: As discussed above, consider including into the three case study sections. L234-236 remains somewhat vague, and would – with some more specific conclusions – fit nicely into Section 4 (Conclusions).

- L240: Have TSE and STE already been introduced? Has STE been used in the sense 'stratosphere-to-troposphere exchange', or did it refer to cross-tropopause transport in both directions`?

- Section 4 (Conclusions): As discussed before, try to bring more specific statements into conclusions.

---

## Author Response (AR1)

**Response to Reviewers**

**April 17, 2025**

We thank the editor and referees for the precious time to review the manuscript and provide valuable comments. Reviewer comments are in **bold**, while our responses are in normal text.

**Response to Reviewer 1**

In section 3.1, the authors compared the model TKE with the turbulence diagnostic TI1 using the grid-scale wind data. Where does this grid-scale wind data come from? Is it data produced by COSMO? And if so, why do not the authors use ERA-5 data to evaluate the ability of COSMO to represent turbulence? Besides, the authors say that the differences between TI1 and TKE "might be caused by the neglected mechanism of the Ellrod index or sub-grid scale processes that could potentially lead to the formation of turbulence in the UTLS, e.g., scale-resolved gravity waves" (lines 137-138). First sub-grid scale processes are mentioned as a possible source of discrepancies and then an example is presented talking about scale-resolved gravity waves. Please, check and clarify this.

Yes, the grid-scale wind data is produced by COSMO. Using the COSMO wind data is because we would like to see whether the model grid scale wind field is consistent with the results of the sub-grid scale turbulent scheme, and considering the simulation is initialized and nudged by the ERA-I data, and the similarity between ERA-5 and ERA-I, therefore we did not choose to use ERA-5 wind field data to evaluate the turbulence. For lines 137-138, we have corrected in the manuscript from "e.g., scale-resolved gravity waves." to e.g., sub-grid scale gravity waves."

Throughout the paper, the authors used different dates and times to analyze turbulence and the distribution of passive tracers for three different cases but did not explain the reasons behind that selection. Could the authors justify why they chose these dates and times? Are they based on aircraft measurements, soundings? Please, explain. The clarity and quality of the paper will improve if information about the synoptic situation is included.

The reason for choosing the selected time for turbulence analysis is that the selected time shows the features consistent with TI1 and features that are potentially caused by the sub-grid scale processes. The three cases are selected to show mixing under different situations. Therefore, they were chosen case by case and did not intend to match with the time of the turbulence analysis. Section 3.3.1 is added to the manuscript to describe the synoptic situation.

In Table 1, the different mixing ratios of the two passive tracers are presented without further explanation about the selection of these values and gradients. Could the authors provide an explanation for these ranges of the mixing values for each of the tracers? Why the different types of gradients for O3 (steep) and N2O(gentle) initialization?

The reasons for the selection of the tracers value and the reason for with different gradients is that we would like to investigate how the gradient could affect the strength of vertical mixing by using passive tracers which have relatively realistic values (which is similar to O3 and N2O) considering the constitutive equation in section 2.3. The values were chosen related to typical atmospheric mixing ratios for future comparisons with real atmospheric data. We have added in the manuscript "An  $O_3$ -like tracer with a relatively steep linear gradient and a  $N_2O$ -like tracer with a relatively gentle gradient are initialised to investigate the effect of the tracer gradient on the strength of mixing under a relatively realistic scenario."

Figures 5 and 6 show the distribution of tracers and the tracer-tracer correlation for a particular cross-section as mentioned in line 154. Where is this cross-section located on the map? Please, include it either in Figure 3 or Figure 4 or explain it in the text. This way the reader knows which section of the map is being analyzed. On a related note, no reference to the location where the three different study cases are performed is indicated in the text. Is it the same location as Figure 5 and 6? Please, clarify.

All 3 cases are located at different parts of the domain, Figure 7 is added in the manuscript to describe the synoptic situation and indicate the location of all 3 cases.

Line 258: here the authors say that the mixing can be irreversible when "the exchange of tracers happens along a diagonal of a delta tracer-tracer correlation" while in the beginning of section 3.3.3 they say that scattered values from the diagonal are an indication of irreversible mixing. Please, clarify.

The scattered value should be related to the advection as well as the tropopause, we have rewrote in the manuscript but more to entries of tropospheric tracers into regions of typically stratospherically dominated regimes in an irreversible way. The scatter away from the diagonal unlike the other 2 cases is most likely due to the advection, considering the completely different wind field in Figure 8 and tropopause in Figure S17, the strong horizontal advection in the region of strong horizontal gradients changes the background ratios in addition to the vertical mixing and thus introduces additional mixing during each time step compared to the other cases. The wider the scatter is, the more, e.g., tropospheric tracer depletion is found at similar stratospheric tracer values , which does not represent a movement of an air mass, but irreversible mixing.

Line 85: could the authors explain why the focus is on the Scandinavian region instead of on the eastern Mediterranean (line 45: the authors say it is an area with strong STE)?

Scandinavian region was selected because it was planned to study gravity waves effects as well in the later stage, which was not included in this manuscript.

Line 88: delete "tbc".

Deleted.

Line 129: "...this section analyses," delete the comma.

Deleted.

Line 131: what do the authors mean by "grid-scale wind data"?

Added in the manuscript "grid-scale wind data from COSMO

Line 154: replace "ooff" by "off".

Corrected.

Figure 5: please include in the caption a description of what the black line represents. It is not explicitly mentioned here or in the text.

Included, the black line represents the PV-tropopause.

Line 187: after "vertical wind shear" please include "(VWS)" so that it is easier for the reader to follow the text while looking at the Figures.

Included

Line 189: please, include a description of N2 as it is the first time that it appears in the text.

Added in the manuscript "the Brunt-Väisälä frequency  $(N^2)$  shows no distinct behavior"

Line 195: add a comma after "Besides the diffusion coefficient".

Added

Please, include the units used in each subplot in Figures 7, 9 and 10.

Included

Line 210: "spheric air could attributed". Change for "could be attributed".

Changed

Line 214: put "the N2 is distinctly higher in this region" inside parenthesis.

Changed

Line 241: delete the comma after "new". Too long sentence. Please, split.

Deleted, Changed from "a new, enhanced vertical resolution model setup (200m vertical resolution in the UTLS) for the regional model COSMO nested within the multi-scale climate chemistry model MECO(n) is presented, which performs similar to ......" to a new enhanced vertical resolution model setup (200m vertical resolution in the UTLS) for the regional model COSMO, which is nested within the multi-scale climate chemistry model MECO(n) is presented. It performs similar to ......"

Line 250. Change the comma after "real world" for a full stop.

Changed

Line 260, replace ".e.g." by ",e.g."
Replaced

**Response to Reviewer 2**

The passive-tracer test in Section 3.2 relies only on one turbulence indicator (Ell-rod and Knapp's TI1), whereas there are many other indicators available (e.g., Richardson number as an indicator for Kelvin-Helmholtz instability, KHI). Hence, the 'validation' against this turbulence indicator must be considered with some caution. Furthermore, the turbulence indicators (TI1, TI2, Richardson number,...) are also not the truth, but may fail in recognizing turbulence and/or indicate turbulence where actually there is none. I suggest that the authors more carefully discuss the limitations (and potential) of the turbulence indicators in the passive-tracer test.

We agree with the reviewer that the TI1 has its limitations and it might neglect or fail to recognize turbulence at some point. The reason for choosing TI1 as an indicator is because Sharman et al. (2006) provides a reference threshold on turbulence strength for TI1 which is not available for some other indicators like TI2 or the improved version of TI1 with a divergence trend term. We agree that including more turbulence indices could provide a more comprehensive validation. Still, it is not the focus of this manuscript so we decided to use one of the well established turbulence diagnostic TI1 in this manuscript. In the ongoing work, we prepared to include a turbulence diagnostic that considers static stability and divergence. We have added in the manuscript in section 3.2 to discuss the limitation of TI1: It is also important to note that the Ellrod index does not fully represent the turbulence in the atmosphere since it does not account for all producing mechanism. For example TI1 might neglect or underestimate the shear related to anticyclonic flow.

The three case studies, particularly the discussing of the delta tracer-tracer correlations are very interesting and inspiring! There are two aspects that might be improved, however. First, Subsection 3.3.1 comes with a short introduction on this type of diagram, which is nice. I think that the reader can even be more clearly be introduced on how to read these diagrams. A few additional sentences might be helpful. For instance, why is an L-shape indicative for asymmetric mixing, whereas a diagonal one is clearly pointing to symmetric mixing; where in these diagrams do we see mixing from stratosphere to the troposphere and vice versa. See also the specific comments below. Second, the three cases in Section 3.3 discuss in detail the mixing and tracer-tracer correlations, which is the key topic of the study. Still, it would also be interesting (at least to me) to learn somewhat more about the dynamical and thermodynamical 'setting' of the cases. For example, how does the wind field look like? How is the vertical stratification? Or, why do we see in Figure 5 mixing at a very distinct location (blue/red dipole) but not at neighboring locations. I fully understand that these aspects are not the key topic of the paper, still some mesoscale context would be interesting.

A few more sentences is added to explain the delta-tracer-tracer correlation. See the response in the specific comment below. A section describing the synoptic condition of the cases is added to the manuscript. The reason for the blue/red dipole is the location of the jet stream and the

tropopause, therefore it experiences strong vertical wind shear and tracer gradients.

A minor point on structure: I wonder whether Section 3.3.4 is actually needed, or whether the aspect of mixing normalization (to determine the importance of gradient vs. dynamically forced mixing) would better integrated in the three case studies. In short, the discussion of this effect is interesting, but maybe not necessarily in an own subsection. In line with this, I wonder whether Figures 11 and 12 are needed in the main text, or could become part of some supplementary material? I have not a very strong opinion on that, but the authors might consider it. Furthermore, the statements in Section 4 (Conclusions) are rather general and vague. Some examples: L245 ("with different characteristics"  $\rightarrow$  which?); L249 ("However, individual mixing events depend on the particular weather situation"  $\rightarrow$  what weather situation? Which ones have been studied in this paper?); L55 ("Both, the vertical tracer gradient and the dynamic and thermodynamic forcing, i.e. the stability and stratification, play important roles in the strength of vertical species exchange"  $\rightarrow$  Maybe, that what we already would have expected! So, can you make a more nuanced statement about the relative importance?).

We added some statements in the manuscript:

L245 "with different characteristics"  $\rightarrow$  including balanced and imbalanced bi-direction mixing induced by turbulence and strong vertical tracer gradient

L249; "on the particular weather situation"  $\rightarrow$ , for example, the vicinity of a jet stream which located near the tropopause (case 1) experiencing the strongest mixing considering the high vertical wind shear and tracer gradient

L255 "Both, the vertical tracer gradient and the dynamic and thermodynamic forcing,i.e. the stability and stratification, play important roles in the strength of vertical species exchange"  $\rightarrow$ , especially when the vertical wind shear is strong enough to overcome the stable atmosphere

L17: aka PV tropopause; introduce 'potential vorticiy' before referring to it as PV

Corrected

L19: 'PV-tropopause is conserved'  $\rightarrow$  I understand what the authors mean: that PV is conserved under adiabatic, frictionless flow conditions, and that thus the PV-tropopause behaves as a material surface under the these assumptions. The authors might rephrase, as 'tropopause is conserved' sounds a little strange.

Rephased in the manuscript from Since the PV-tropopause is conserved under isentropic conditions to Since the PV-tropopause is a quasi-impermeable surface for adiabatic frictionless flow, i.e., on isentropes.

L27: affecting  $\rightarrow$  affects

reply

Figure 1: Are really both panels needed?

For the first panel, we would like to show the location of the domain on the globe considering not every reader is familiar with the Scandinavian geography. The second panel we would like to emphasize the irregular shape of the domain with respect to the parallels and meridians.

L88:  $COSMO(tbc) \rightarrow What does 'tbc' mean?$

Typo. Deleted

L95-98: Here, first the default setup of the 1D closure is described, whereas in the following text the actually used TKE-based scheme is described. Is it really necessary to describe the default scheme when it is not used? I see the authors' point to highlight in which way the default setup has been improved, but still wonder whether it is needed...

We try to let the reader know what is the difference between a default setup and the setup that we have chosen and also to show the limitation of the default scheme and it is the reason that we chose the other one.

L103-106: As before, do we need such a detailed description of the default setting? Or would be sufficient to just highlight in the text the improved setup?

We tried to point out that the default setup is insufficient for UTLS research which motivates the implementation of the enhanced vertical grid.

L116: Very minor detail: I think it should be ERA5 instead of ERA-5; and possibly start a new paragraph after 'EX-60 setup'?

Corrected and started a new paragraph.

Section 3.1 (as discussed above): To which degree does this analysis depend on the specific section of the TI1 turbulence indicator? To which degree can the TI1 threshold to be assumed independent from the model resolution? Would you expect different results if TI2 and/or Richardson number is used instead of TI1? No detailed analysis is needed in my opinion, but a more critical discussion of the comparison betweem TI1 and the model TKE.

We would expect the TI2 shows a similar results as the TI1 with minor discrepancy since both TI are based on deformation and vertical wind shear, but we would expect discrepancy arise from the convergence term of TI2, We have added in the manuscript in section 3.2 to discuss the limitation of TI1: It is also important to note that the Ellrod index does not fully represent the turbulence in the atmosphere since it do not account for all producing mechanism. For example, TI1 might neglect the shear related to anticyclonic flow [Ellrod and Knox, 2010].

Figure 5 (and corresponding text): I like the discussion on the tracer and inverted-tracer mixing, but wonder as a dynamic meteorologist why the mixing only occur where it does and not at the other locations. Some meteorological background/context for the mixing events would be interesting, possibly supported by figures in the supplementary material?

Section 3.3.1 describing the synoptic situation is added in the manuscript. The mixing region occurs in the vicinity of the jet stream with strong tracer gradient.

Figure 6 (panels c an d): Whereas panels a) and b) show straight diagonals, panels c) and d) exhibit some off-diagonal deviations. I wonder whether I correctly understand these. As an example, in panel c) in the lower-left quarter O3 with vdiff corresponds to somewhat belowdiagonal O3 without vdiff. Does this mean that mixing (with vdiff) lowers the O3 values?

No, It should be the other way around. O3 below the diagonal means there are more O3 with

vidff and above the diagonal mean less O3 with vidff. Take the lower left quarter as an example. the O3 with vdiff have a value larger than 1e-6 while the O3 without vdiff have a value smaller 1e-6, indicating mixing with vdiff increase the O3 value. Note that in the upper right quarter, the relation is opposite. This is due to the background gradient. Increased mixing redistributes the tracer and tends to homogenize gradients, leading to a dipole effect (See Fig. 5).

Introductory paragraph to Section 3.3.1: Possibly, introduce the reader even more carefully how to read the delta tracer-tracer plots (as explained above). Why does an L-shale indicate asymmetric mixing? Where in the plots do we see mixing from the stratosphere into the troposphere, where in the opposite direction? Somewhat reflecting on it, it becomes obvious? But why not helping the reader not familiar with tracer-tracer plots with 1-2 explaining sentences to get it at first reading?!

We have add some sentences in the manuscript from (1) Concentrated distribution at the center [0,0] if no vertical mixing takes place at all;(2) L-shape distribution for singledirectional mixing; and (3) Diagonal distribution for bi-directional mixing to (1) Concentrated distribution at the center [0,0] if no vertical mixing takes place at all; (2) Diagonal distribution for bi-directional mixing, where both tracers changes in a similar rates causing the data point spread along the diagonal. The upper left indicates the downward mixing of stratospheric air into the troposphere since at the same grid, there are increasing stratospheric tracer and decreasing tropospheric tracer. And the lower right indicates the opposite, with decreasing stratospheric air and increasing tropospheric air i.e. upward mixing of the tropospheric air.

**L181: Minor detail; maybe start new paragraph**

Changed

L 185: 'by dynamic instability'  $\rightarrow$  The term 'dynamic instability' is somewhat vague/unclear? I suggest that the authors introduce at some point in the manuscript very clearly what they mean with 'dynamic instability', and also set into contrast to other instabilities? Hence, if it is not a dynamic instability, what is it then? I assume that 'dynamic' refers essentially to the ones where wind (horizontal and vertical shear/deformation) is important, whereas a 'static instability' refers to one where a vertical profile becomes (thermo-dynamically) becomes unstable (e.g., if squared BruntVäisällä frequency becomes negative)? In essence, I suggest to carefully introduce these terms...

Introduced in the manuscript from caused by dynamic instability to caused by vertical wind shear or/and deformation

L199: I get the basic idea, but what does |mixing| and |mixing| / |gradient| exactly stand for?

They are the absolute value of the difference with and without vdiff for the O3-like passive tracer, reworte in the manuscript from tracer-tracer plot but color-coded with |mixing| (left) and |mixing| / |gradient| (right). to tracer-tracer plot but color-coded with absolute value of the difference of the stratospheic O3-like tracer (left, |dO3ST|) and absolute value normalized with the tracer gradient (right, |dO3ST| / |gradient|).

L206-207: "This is a consequence of asymmetric strength and extension of the dynamical forcing, i.e., the stable layering of the stratosphere prevents deeper mixing into the stratosphere, whereas the lower static stability in the troposphere

allows for deeper penetration of stratospheric tracers into the troposphere"  $\rightarrow$  I am not sure whether I fully understand this argument. First, what exactly is 'dynamical forcing' referring to? The discussion in the 'i.e.' sentence is more about stability than on dynamics. Second, the asymmetry argument would also be true in case 1?! Hence, also there the stably stratified stratosphere inhibits a deeper mixing than the more weakly stratified troposphere? Maybe, the solution to this is in the comparison of Figure 9d) with Figure 10d), which shows squared N and clearly differs between the two? In short, some further explanations might help (1-2sentences).

The dynamical forcing is referring to the vertical wind shear and deformation. We are trying to point out that the differences (Case 1 have a similar spread from zero to the higher mixing value on both ends, while Case 2, the lower right quarter is significantly shorter than the upper left quarter.) between Case 1 and 2 is because of the N2 (the lower right of Case 2 are the region with high N2 value). We have changed in the manuscript from This is a consequence of asymmetric strength and extension of the dynamical forcing to This is a consequence of asymmetric stability and flow conditions and added The lower right have a significantly shorter range than the upper left (unlike Case 1, which has a similar range on both ends). and (where in figure 9d, the lower right with high  $N^2$  have a shorter range than the upper left with low  $N^2$ , while figure 7d of case 1 have a similar  $N^2$  on both ends)

L218-221: Please explain in greater detail how the scatter from the diagonal relates to irreversible mixing. I think I kind of get the basic idea, but some further details could be helpful.

The scatter from the diagonal is related to the unusual wind field and tropopause. We have modified in the manuscript but more to entries of tropospheric tracers into regions of typically stratospherically dominated regimes in an irreversible way. The scatter away from the diagonal unlike the other 2 cases is most likely due to the advection, considering the completely different wind field in Figure 8 and tropopause in Figure S17, the strong horizontal advection in the region of strong horizontal gradients changes the background ratios in addition to the vertical mixing and thus introduces additional mixing during each time step compared to the other cases. The wider the scatter is, the more, e.g., tropospheric tracer depletion is found at similar stratospheric values , which does not represent a movement of an air mass, but irreversible mixing.

Section 3.3.4: As discussed above, consider including into the three case study sections. L234-236 remains somewhat vague, and would – with some more specific conclusions – fit nicely into Section 4 (Conclusions).

We have added in Section 3.3.4, The tracer gradient plays the most important role since mixing will be meaningless if there is no tracer gradient. Strong dynamical forcing like vertical wind shear could lead to mixing even in the stable atmosphere with a typical stratospheric  $N^2$  value.

L240: Have TSE and STE already been introduced? Has STE been used in the sense 'stratosphereto-troposphere exchange', or did it refer to cross-tropopause transport in both directions'?

We thank the reviewer for this point, TSE is not defined. It refers to cross-tropopause transport in both directions in the introduction. We clarified the sentence from quantification of TSE and STE with implications on the Earth's radiation budget. to quantification of the bi-directional cross-tropopause transport with implications on the Earth's radiation budget.

Section 4 (Conclusions): As discussed before, try to bring more specific statements into conclusions.

Added in the conclusion including balanced and imbalanced bi-direction mixing induced by turbulence and strong vertical tracer gradient and, for example, the vicinity of a jet stream which located near the tropopause (case 1) experiencing the strongest mixing considering the high vertical wind shear and tracer gradient

**Response to Reviewer 3**

Model Validation: The authors utilize a multi-scale model framework with increased vertical resolution to represent mixing events due to Clear-Air Turbulence (CAT). However, I do not find any validation of this approach that ensures its suitability for studying CAT effects on mixing. True, the authors compare the Turbulent Kinetic Energy (TKE) produced by the turbulence scheme with a turbulence index, but this index is computed from the same model data and contains terms (such as the shear term) that enter the TKE equation itself. How does the predicted TKE compare with observations? Why should the proposed resolution (horizontal and vertical) be sufficient to study turbulent exchange in the UTLS? To address this question to some extent, the authors can compare the simulated TKE in the CM10 and CM40 models. The limitations of the present approach should be stated clearly in the introduction and conclusions, especially considering the stated objective (line 55) of analyzing the representation and efficiency of turbulent tracer mixing in the UTLS.

The reason for comparing the TKE with TI1 is to see whether the highly parametrized subgrid scale turbulence scheme is consistent with the grid-scale wind. A sensitivity study was conducted on the supplement to show the necessity for higher vertical and horizontal resolution. Figure S9 shows the relative frequency distribution of the model TKE between EH84 and EX60, and the occurrence of high TKE values is more frequent in EH84 than in EX60. The difference caused by vdiff is also an order of magnitude stronger between EH84(figure 5 in manuscript) and EX60 (figure S11). For CM10 and CM40, they show similar the frequency distribution (figure S10), but the difference caused by vdiff is still significantly stronger between CM10 (figure 5 in manuscript) and CM40 (figure S12). we have included some more discussion in the section of turbulence analysis (section 3.1) It is also important to note that the Ellrod index does not fully represent the turbulence in the atmosphere since it do not account for all producing mechanisms. For example, TI1 might neglect the shear related to anticyclonic flow.

Sponge Layer Depth: Another critical point in this study is the tiny depth of the sponge layer (5 km), which is significantly less than the 11 km used in the CM40 and CM10 configurations. Typically, the sponge layer accounts for about half of the vertical domain extent. The authors should comment or provide a reference explaining why 5 km is a reasonable choice for the high-top model configuration. Additionally, how do they ensure that wave reflections from the model top do not affect the UTLS region?

The 5 km sponge layer is chosen according to [Eckstein et al., 2015], which validates a setup extending the COSMO vertical grid reaching the lower stratosphere. They also tested a sponge layer with 11 km, and the differences were marginally small. We also see similar results between

an 11 km and 5 km damping layer when we test the enhanced grid (EH-84). Therefore, we stick with the 5 km sponge layer. On the other hand, the analysis carried out in this manuscript is far below the damping layer, at below 15 km, with 20 model levels until it reaches the model top. The potential reflections from the model top should be negligible.

Simulation Setup: Any motivation for the choice of the selected geographic and temporal window for the simulation is missing. What is the large-scale configuration of meteorological fields, and why is it expected to be favorable for Stratosphere-Troposphere Exchange (STE)? The authors show a high turbulence index over the selected region (Fig. 3), but they do not discuss a possible underlying cause. Is the elevated turbulence index due to shear in the jet stream, is it caused by gravity waves or something else?

Large-scale climate models also use the TKE equation, but the simulated shear is often too low. If the authors comment on the processes responsible for the increased shear in their high-resolution simulation, it would be valuable for readers working on improving large-scale climate models.

The choice of the selected geographic and temporal window is based on a previous airborne measurement campaign that took place at Kiruna. However, the measurement data had not been used in this manuscript and therefore was not mentioned in the manuscript. A section is added in the manuscript to show the synoptic condition and explain the choice briefly. We expected that the high-level trough ridge system would be favorable for CAT, which has a potential impact on STE. Yes, the elevated turbulence index is due to the jet stream, as shown in Figure 1 which shows the horizontal wind speed at 200 hPa. The increased shear could be attributed to the higher vertical and horizontal resolution, as in Figure 2, the CM10 shows more fine structures and hence more shear.

Figure 1: Horizontal wind speed at 200 hPa for CM10

Figure 2: Cross section for horizontal wind speed for CM10 and EMAC

Missing Simulation Details: Important details regarding the simulation from Sec. 3 are missing: What are the initial conditions used for the tracer? It is stated that the tracer gradients are initially linear, but the gradients in Fig. 5 are far from linear. How long does it take to reach the state shown in Fig. 5? When was vertical diffusion switched off—at the very beginning of the simulation, or only just before the particular event.

The vertical diffusion was switched off at the very beginning, the submodel PTRAC of MESSy allows the user to switch on and off certain processes for certain tracers. For each tracer two versions with identical initialization exist, one with vertical diffusin (vdiff) active and one without, the detail is listed in Table 1. The below figure shows the initial condition of one of the tracers, all passive tracers were initialized similarly except for the range of the mixing ratio. In general, it takes only a few days for the tracer to reach a similar state as figure 5, since we initialized the tracer with a transition layer near the tropopause.

Figure 3: Initial condition of the N2O-like tracer

Analysis of Mixing and Exchange: Tracer-tracer correlations are used to discuss the direction of troposphere-stratosphere exchange. While this is a valuable diagnostic in observational studies, given the model data, is it not possible to diagnose the diffusive fluxes directly (using the equation on page 4) and confirm the findings from the tracer-tracer correlations?

Additionally, it is not immediately clear to the reader how the mixing direction can be inferred from a delta tracer-tracer correlation. For example, at line 209, stronger downward mixing is concluded from Fig. 9. A more extensive explanation of the interpretation of the novel delta tracer-tracer correlation would be helpful here.

Yes, the fluxes of the tracer by vertical diffusion could be derived by the model, we do see fluxes locally near the mixing region of e.g. figure 5, but what we see in figure 5 is an accumulation of the fluxes. However, the fluxes would be difficult to compare with the tracer-tracer correlation since it also shows an accumulation of tracer due to advection of mixed air from other events. For the delta tracer-tracer correlation, we have added some more explanation in Section 3.3.2 in the introduction of the delta tracer-tracer correlation.

**Fig. 5: What is the longitude of this cross-section? Can you indicate it in Fig. 4? It would be better to present in Fig. 3 and Fig. 4. results corresponding to the event from Fig. 5**

The longitude of the cross-section is at 18°. The cross-section is now indicated in figure X in the new section describing the synoptic conditions.

**Figures 8, 11, and 12: How is mixing defined in these figures? What are the units [dO3ST]?**

They are the absolute value of the difference with and without vdiff for the O3-like passive tracer, reworte in the manuscript from tracer-tracer plot but color-coded with |mixing| (left) and |mixing| / |gradient| (right). to tracer-tracer plot but color-coded with absolute value of the difference of the stratospheic O3-like tracer (left, |dO3ST|) and absolute value normalized with the tracer gradient (right, |dO3ST| / |gradient|). The units of dO3ST is  $(kg^{-1}kg^{-1})$ , added in the manuscript.

Figures 9 and 10: These figures show different mixing regimes compared with the balanced bi-directional mixing from case 1. It would be informative to show the corresponding vertical cross-sections of the tracer and tracer difference (vdiff onoff) as in Fig. 5.

Cross section is now included in the supplement (Figure S16 and S17).

Section 3.3.3: The introduction states that turbulence provides irreversible mixing, but here only the scatter in Fig. 10 is associated with irreversible mixing. This section should be rewritten for clarity, and the discussion and conclusions from Fig. 10 should be more precise.

All the cases are associated with irreversible mixing, considering the composition of the air masses is substantially modified, and the tracer is mixed irreversibly into the grid. We have clarified it in the manuscript to but more to entries of tropospheric tracers into regions of typically stratospherically dominated regimes in an irreversible way. The scatter away from the diagonal unlike the other 2 cases is most likely due to the advection, considering the completely different wind field in Figure 8 and tropopause in Figure S17. The wider the scatter is, the

more, e.g., tropospheric tracer depletion is found at similar stratospheric values , which does not represent a movement of an air mass, but irreversible mixing.

Figure 9: Is the correct time 05:00 or 04:00 as stated in the text?

Corrected. The correct time should be 0500.

Line 154: Change "vdiff oof" to "vdiff off."

Corrected

**References**

J. Eckstein, S. Schmitz, and R. Ruhnke. Reaching the lower stratosphere: validating an extended vertical grid for cosmo. *Geoscientific Model Development*, 8(6):1839–1855, 2015. doi: 10.5194/gmd-8-1839-2015. URL https://gmd.copernicus.org/articles/8/1839/2015/.

Gary P. Ellrod and John A. Knox. Improvements to an operational clear-air turbulence diagnostic index by addition of a divergence trend term. Weather and Forecasting, 25(2):789 – 798, 2010. doi: 10.1175/2009WAF2222290.1. URL https://journals.ametsoc.org/view/journals/wefo/25/2/2009waf2222290\_1.xml.

---

## Referee Report (RR1)

**Second Review of the Paper:** "Simulated Mixing in the UTLS by Small-Scale Turbulence Using Multi-Scale Chemistry-Climate Model MECO(n)" by Chun Hang Chau, Peter Hoor, and Holger Tost**

The revised version of the manuscript does not sufficiently address the major comments raised in the initial review. In many cases, the authors' responses are not reflected in the revised text, or only marginally so. I strongly recommend that the authors carefully read all reviewer comments, revise the manuscript accordingly, and ensure that the responses are clearly visible and well-integrated into the main text.

**Model Validation**

The response to Comment 1 is not adequately incorporated into the revised manuscript. Aside from a brief addition in Section 3.1, no substantial changes are evident. Furthermore, the limitations of the modeling approach -- explicitly raised in the original review -- are still not clearly discussed in either the Introduction or the Conclusions.

**Sponge Layer**

The revised manuscript should include a clear comment on the validation of the sponge layer depth, as provided in the authors' reply. Currently, this is missing.

**Simulation Setup**

Figures 1 and 2 presented in the reply contain important information but are not discussed or included in the revised manuscript. The paper should be rewritten to incorporate and discuss these figures.

**Missing Simulation Details**

It remains unclear why the details provided in the authors' reply were not integrated into the revision. In addition, instead of the 2D plot shown in Fig. 3, a vertical profile will be sufficient. Key information is still lacking -- regarding the width of the transition layer.

**Analysis of Mixing and Exchange**

In their reply, the authors state:

"The fluxes would be difficult to compare with the tracer—tracer correlation since it shows an accumulation of tracer due to advection of mixed air from other events."

If this is indeed the case, it raises fundamental question:

- Why are individual events analyzed at all, if they cannot be clearly separated?
- Why is this critical limitation not explained in the manuscript?

---

## Referee Report (RR2)

**3. Review of the Paper: "Simulated Mixing in the UTLS by Small-Scale Turbulence Using Multi-Scale Chemistry-Climate Model MECO(n)" by Chun Hang Chau, Peter Hoor, and Holger Tost**

The authors have made an effort to implement the major comments, and I think the readability of the paper has improved as a result. However, I still have a question regarding the final comment about the *Analysis of Mixing and Exchange*. I understand that the delta tracer-tracer correlation is not suffering from the limitations of the traditional tracer-tracer correlation related to accumulation effects. Setting aside the issue of the method's applicability - since two experiments have to be performed to find the delta tracer-tracer correlation - the authors might address the issue raised in question 5 of the original review: Can the fluxes be computed directly from the simulation data and used to support the findings not from the tracer-tracer but from delta tracer-tracer correlations.

---

## Author Response (AR2)

**Response to Reviewers**

May 28, 2025

We thank the editor and referees for pointing out that some additional information should be included for the wider audience. Reviewer comments are in **bold**, while our responses are in normal text.

**Response to Reviewer 3**

Model Validation: The response to Comment 1 is not adequately incorporated into the revised manuscript. Aside from a brief addition in Section 3.1, no substantial changes are evident. Furthermore, the limitations of the modeling approach – explicitly raised in the original review – are still not clearly discussed in either the Introduction or the Conclusions.

We added in the manuscript "Furthermore, a sensitivity study was conducted on the supplement to show the necessity for higher vertical and horizontal resolution. The occurrence of high TKE values in EH-84 is more frequent than in EX60 (Figure S9), hence the mixing (Figure 5 and S11). CM10 (Figure 5) also shows more efficient mixing than CM40 (Figure S12) despite the similar TKE frequency."

We also added in the manuscript "the model turbulence kinetic energy (TKE) is compared with a calculated turbulence index using the grid-scale wind data from COSMO, i.e., the turbulence diagnostic TI1 from Ellrod and Knapp (1992), which includes a vertical wind shear term and a deformation (stretching and shearing) term to examine whether the highly parametrized subgrid scale turbulence scheme is consistent with the grid-scale wind."

We also added "It is also important to note that the comparison between the TI1 and TKE is not sufficient enough considering both of them are calculated from the COSMO wind field; comparison with observation should be conducted when there is available data."

We also added in the conclusion"However, further comparison with observation is needed considering both of them are calculated from the COSMO wind field."

Sponge Layer: The revised manuscript should include a clear comment on the validation of the sponge layer depth, as provided in the authors' reply. Currently, this is missing.

We added into the manuscript "Our enhanced setting has 84 levels and reaches also up to 33 km, with an identical 5 km damping layer starting at 27 km considering Eckstein et al. (2015) shows that the differences between a 5 km and 11 km damping layer is marginally small, and the analysis carried out in this study are far below the damping layer with more than 20 model levels, the potential reflection from the model top should be negligible."

Simulation Setup: Figures 1 and 2 presented in the reply contain important information but are not discussed or included in the revised manuscript. The paper

should be rewritten to incorporate and discuss these figures.

Section 3.3.1 is added to describe the large-scale meteorology. We also added in the manuscript "The high TKE and Ellrod index shown in Figure 3 and Figure 4 is caused by the jet stream as shown in Figure S18. The increased shear could be attributed to the higher vertical and horizontal resolution, as shown in figure S19. CM10 shows more fine structure and hence more shear." In order not to make the manuscript too lengthy, we have added figures 1 and 2 of the horizontal wind from the first reply into the supplement as figures S18 and S19.

Missing Simulation Details: It remains unclear why the details provided in the authors' reply were not integrated into the revision. In addition, instead of the 2D plot shown in Fig. 3, a vertical profile will be sufficient. Key information is still lacking – regarding the width of the transition layer.

We added in the manuscript "For the same pair of tracers, the only difference is whether the physical process of vertical diffusion (vdiff) is turned on or off. The vertical diffusion was switched off at the very beginning." We also added in the manuscript "The tracers change linearly at the transition layer near the tropopause between approximately 300 hPa to 150 hPa. The initial condition of the tracers is shown in figure S20 as a vertical profile." Again, we added the vertical profile into the supplement as figure S20 to ensure the main manuscript isn't overly lengthy.

Analysis of Mixing and Exchange: In their reply, the authors state: "The fluxes would be difficult to compare with the tracer—tracer correlation since it shows an accumulation of tracer due to advection of mixed air from other events." If this is indeed the case, it raises fundamental question:

- Why are individual events analyzed at all, if they cannot be clearly separated?
- Why is this critical limitation not explained in the manuscript?

In the original review, the reviewer mentioned "Tracer-tracer correlations are used to discuss the direction of troposphere-stratosphere exchange. While this is a valuable diagnostic in observational studies, given the model data, is it not possible to diagnose the diffusive fluxes directly (using the equation on page 4) and confirm the findings from the tracer-tracer correlations?" Tracer-tracer correlations are not used to discuss the direction of troposphere-stratosphere exchange, instead, the delta tracer-tracer correlation is used. The tracer-tracer correlation cannot derive the mixing direction or the mixing from a single event, it shows a background accumulation of the mixing from all kinds of sources over time, therefore, it is not possible to directly compare with the diffusive fluxes from the equation on page 4. Therefore, we introduced the delta tracer-tracer correlation, which can show in the model the direction of the mixing. Since we can switch off the vertical diffusion in the model, we show in cross-section like figure 5 the mixing by the turbulence vertical diffusion only which can be directly related to the delta tracer-tracer correlation. Considering the tracer field of COSMO is initialised and boundary driven from EMAC, which is without an efficient turbulence scheme in the free troposphere, the differences in COSMO originate from turbulent mixing after the tracer enters the domain showing the accumulation of the mixing in the domain before the airmass propagates out of the domain. Consequently, the mixing in the cross-section must be the accumulation of the mixing within several hours, most likely from the same events or at least affected by the same meteorological condition, otherwise it is out of the domain already. So in principle, delta tracertracer correlation is trying to show a similar concept as the tracer-tracer correlation, but not an accumulation of all, only the accumulation of a single process. We also added in the manuscript "It is a similar concept as the tracer-tracer correlation, but instead of showing the mixing as an

accumulation of all related processes, it shows the impact of a single process."

**Response to Reviewer 2**

I wonder whether it would be worthwhile to include somewhat more information, e.g., in the vertical cross sections of horizontal wind. For example, one could show there: the location of the tropopause, some TI1 contour lines, some contour lines from Figure 5c,d of the ozone tracers,... In short, now the wind cross-sections stand quite 'isolated', also the discussion in the text does not go beyond stating whether the cases are or are not near the jet. A few more sentences would be nice that relate more directly the ozone mixing to the jets.

We add the O3-like tracer contour line in the vertical cross sections of the horizontal wind. We have already added in the section describing the synoptic that "Case 1 and 2 (figure 8a and 8b) are also associated with the jet stream, while in case 3 (figure 8c) the jet stream was shifted outside of the CM10 model domain."

---

## Author Response (AR3)

**Response to Reviewers**

August 29, 2025

We thank the editor and referees for the comments. Reviewer comments are in **bold**, while our responses are in normal text.

**Response to Reviewer 3**

The authors have made an effort to implement the major comments, and I think the readability of the paper has improved as a result. However, I still have a question regarding the final comment about the Analysis of Mixing and Exchange. I understand that the delta tracer-tracer correlation is not suffering from the limitations of the traditional tracer-tracer correlation related to accumulation effects. Setting aside the issue of the method's applicability - since two experiments have to be performed to find the delta tracer-tracer correlation - the authors might address the issue raised in question 5 of the original review: Can the fluxes be computed directly from the simulation data and used to support the findings not from the tracer-tracer but from delta tracer-tracer correlations.

As mentioned in the previous response, the delta tracer-tracer correlation shows the accumulation of a single process on the tracers, while the fluxes calculated by the COSMO turbulence scheme show the instantaneous mixing, at that time step. First of all, it will be impossible to output and store all of the fluxes at every time step. Second, it will most likely show something similar to the TKE or the mixing coefficient, basically showing the region where mixing happened. The delta tracer-tracer correlation or the cross-section of the difference plot (figure 8) shows the consequence/accumulation of the mixing, taking the meteorological conditions into account. Unless we are in an ideal situation where only vertical mixing will modify the tracer distribution (without any horizontal advection from previous potential vertical mixing upstream), we can directly use the fluxes to compute the tracer mixing. Otherwise, the fluxes cannot directly determine the tracer mixing.

**Response to Editor**

1) Motivation and scientific question - As it comes now, the manuscript does not convey a sufficiently new or important message. The importance of UTLS composition, of the role of turbulent mixing in shaping it and of the issue of how well high-resolution simulations can capture it, is beyond any question. However, after reading your manuscript I am struggling to disentangle the motivation and goal of your manuscript. Was the aim to produce a detailed case study of turbulent events above Scandinavia? To test the skill of high-resolution simulations for turbulent mixing in UTLS? To test the performance of the parameteriation and get more insight in how the turbulence scheme by Doms et al works? Or to introduce a

**novel delta tracer methodology for analysing the mixing in the models?**

The key focus of our study is the representation of tracer mixing in the UTLS by turbulence in relatively coarse grain models. We conducted a systematic analysis of the impact of turbulence on tracer mixing in model simulations, which, to our knowledge, has never been published in such a detailed way. Most of the literature focus on turbulence in a dynamical point of view, but not the impact of the turbulence on tracer mixing. We are not trying to analyse the turbulence representation itself but rather the representation of turbulence and related tracer mixing. We analyse if the tracer mixing can be either deduced from the tracer gradient or other related forcing. We reformulate in the introduction from

Considering the increasing trend of CAT, and the link between turbulent mixing and STE, and hence the radiation budget, it is crucial to investigate the relation between CAT and mixing of chemicals in the UTLS. The main objective of this study is to analyse the representation and the efficiency of turbulent tracer mixing in the UTLS utilising the multi-scale climate chemistry model MECO(n).

to

Considering the increasing trend of CAT, and the link between turbulent mixing and STE, and hence the radiation budget, it is crucial to investigate the relation between CAT and mixing of chemicals in the UTLS. However, previous studies mainly focus on the dynamical aspect of turbulence (Kaluza et al., 2021; Muñoz-Esparza et al., 2020), but not on tracers. The main objective of this study is not to analyse the representation and strength of the turbulence itself, but to systematically analyse the impact of turbulence on tracer mixing in the UTLS. For that purpose, a novel diagnostic, namely the delta tracer-tracer correlation is used within the multiscale climate chemistry model MECO(n). Consequently, the main objective of the study is on the resulting effects on the tracer distributions caused by turbulent mixing. Note to differentiate between the mixing itself, i.e., the "dynamical" mixing represented by e.g., the TKE, and the local effects on the tracer distributions provided by the mixing and the tracer gradient and further effects of the mixing, i.e., the downwind changes in the tracer distributions, originating from the mixing and subsequent processes, e.g., advection. Especially, the latter can further enhance vertical differences in tracer concentrations in case of modified vertical gradients of the respective tracers.

2) Do I understand correctly that you are interested only in the part of mixing that is unresolved by the model and is mediated by the Doms et al. (2018) parameterization scheme? If so, the scheme needs to be detailly introduced in the paper, including all the underlying physics and technical details and review of previous validation efforts of the scheme.

We are interested in how the parameterized turbulence by the Doms et al. (2018) impacts the tracer distribution. However, since we do not have any new model development on the scheme, it will be inappropriate to include the detailed description of the scheme in the manuscript. We have already briefly introduced what related mechanism the scheme is based on. The turbulence scheme is originally designed for and well-tested in the boundary layer. It is known that in turbulence schemes in other model like ECHAM5, which have originally been designed for the boundary layer, the turbulence is dampened in the free atmosphere, it does not change the tracer distributions significantly. Therefore, we would like to address how the COSMO turbulence scheme works on tracer mixing in the UTLS. Nevertheless, we extend the description of the used turbulence scheme from:

COSMO also provides another newer turbulent scheme based on prognostic turbulent kinetic energy. The  $K_{\psi}$  in this prognostic TKE-based scheme is determined by the Blackadar length scale, stability functions and the turbulent velocity scale which is based on the prognostic TKE

equation. The latter scheme is used in this study. Details for the turbulent schemes can be found in the documentation of the COSMO model by Doms et al. (2018).

COSMO also provides another newer turbulent scheme based on prognostic turbulent kinetic energy. The  $K_{\psi}$  in this prognostic TKE-based scheme is in the form of:

$$K^H = q\lambda S^H$$

$$K^M = q\lambda S^M$$

where  $K^H$  and  $K^M$  are the turbulent diffusion coefficients for heat and momentum respectively. They are computed by the corresponding stability functions for scalars  $(S^H)$  and for momentum  $(S^M)$  which are determined by the flux-Richardson number, the turbulent length scale  $\lambda$  (which is assumed to be the Blackadar mixing length) and the turbulent velocity scale  $q = \sqrt{2\overline{e_t}}$  where  $\overline{e_t}$  is the turbulent kinetic energy (TKE). The latter scheme is used in this study. Details for the turbulent schemes can be found in section 3 (3.3.2 for the used scheme) of the COSMO model documentation by Doms et al. (2018).

3) Thorough meteorological analysis of the case studies needs to be done, including identification of regions of dynamic (e.g. Ri) or static instabilities (e.g. using potential temperature contours).

We now added the Ri plots next to the horizontal wind plot as requested by the previous reviewer. The dynamics like vertical wind shear or static stabilities should be able to be derived from the color code of the scatter plot as well. We have rearranged the manuscript; instead of having a specific section for the synoptic situation, we merged it into every case study to improve the readability of the manuscript. The original section from L170–L190 also merge into the case study of case 1.

4) The methodology of computation of tracer-tracer and delta tracer correlations needs to be clarified, because right now it is not clear how they were constructed from the model fields (e.g. sampled over fixed volume regions over time of the simulation?)

We have now added a new section Delta tracer-tracer correlation to explain the concept of it. We added in the manuscript In order to investigate the tracer mixing in the UTLS, we introduced a novel diagnostic, namely a delta tracer-tracer correlation, which is a similar concept to the tracer-tracer correlation, but makes use of the model capabilities. While the tracer-tracer correlation can be compared to the real world, the delta tracer-tracer correlation is a correlation between the differences of the tracers from model experiments. Instead of showing the mixing as an accumulation affected by other processes, it shows the impact of a single process (and potential subsequent advection differences). It requires 2 pairs of tracers (one pair of stratospheric and one pair of tropospheric). The difference of each pair, is a particular process being deactivated on one of the tracers in the model to investigate the impact of it. In our study, it is the turbulent vertical diffusion (vdiff). The detailed released tracers are described in section 3.2. The delta tracer-tracer correlation can also be used to determine the direction of vertical mixing. Several distributions are expected for different scenarios: (1) Concentrated distribution at the center [0,0] if no vertical mixing takes place at all; (2) Diagonal distribution for bi-directional mixing, where both tracers change at a similar rate, causing the data point spread along the diagonal. The bi-directional mixing could be either balanced or imbalanced, meaning an even (case 1, spread equally from the center [0,0]) or uneven (case 2, spread unequally from the center [0,0]) spread along the diagonal. Balanced bi-directional mixing indicates a similar amount of stratospheric tracers being exchanged with the tropospheric tracers, while imbalanced bidirectional mixing indicates a different amount of stratospheric tracers being exchanged with the tropospheric tracers. It could be attributed to different situations, details are discussed in the following cases. The upper left section of the diagram indicates the downward mixing of stratospheric tracers into the troposphere since at the same grid, there are increasing stratospheric tracers and decreasing tropospheric tracers. And the lower right quadrant indicates the opposite, with decreasing stratospheric tracers and increasing tropospheric tracers i.e. upward mixing of the tropospheric tracers. Scatter further away from the center indicates irreversible mixing, as the composition of the air masses is substantially modified, and the tracer is mixed irreversibly into the grid, i.e., instantaneously horizontally mixed. Additionally, this scatter is caused by initial differences from the mixing which are then amplified by (mostly horizontal) advection into regions where the vertical gradient of the tracers are different. Those different gradients can originate both from the tracer mixing event itself further upstream or from specific meteorological conditions, e.g., tropopause folds with strong gradients. Scatter away from the diagonal (case 3) indicates that the mixing occurs in a region with a different tracer gradient, a non-local effect introduced by other processes like horizontal advection acted on the mixed tracer. The scatter away from the diagonal gives an indicator that the mixing is non-local but the strength of mixing itself is still solely contributed by the turbulent mixing.

For the delta tracer-tracer plot in the manuscript, we plot every grid point between 100 to 350 hPa on the cross section that is indicated in Fig. 6, 12,15 (the plot of geopotential height). In the updated manuscript we have now separated the synoptic section into each case with the location of the analysed case and added in the manuscript It is conducted using every grid point between 100 hPa to 350 hPa at the indicated location on Figure 6.to emphasize it. However, this diagnostic should theoretically be working for a different selection of the data, as long as the deactivated process is modifying the tracer in that dataset.

5) Section 3.1 does not have any good meaning and should be omitted from the revision. Definitely, the analysis presented cannot be used for the argumentation (however vague):" To conclude, the model is able to represent turbulence at a reasonable position (and time)." Or was the intention to test the skill of the Ellrod index in detecting the turbulence in the model?

We would like to keep the comparison between TI and TKE to show the consistency between grid scale wind field and the turbulence scheme. To test whether TI is able to detect turbulence in the model is part of the purpose. We have now also added a comparison with measurements to show the link between the model TKE and observation. We have now added in the manuscript We also compare the model results with the last flight in the GW-LCYCLE II campaign (Witschas et al., 2023) on the 1st of February in northern Scandinavia. We derive a measure of turbulence from the high frequency measured N2O (Lachnitt et al., 2023) and link it with the model TKE. We computed a 31-point running standard deviation normalized with the variability of the window for  $N_2O$ . The running standard deviation shows the  $N_2O$  fluctuation from the background in a short period of time while the normalization eliminates the effect of a tracer gradient due to the changing flight altitude or large-scale exchange of air masses. Figure 5 shows the model TKE at the flight time with the normalized running standard deviation of the measured N2O. It shows that the derived turbulence signal often coincides well with the simulated TKE (figure 5a); the stronger signals (higher percentiles, figure 5b) coincide with the higher model TKE as well. This indicates that there is a reasonable degree of consistency between the derived turbulence signal from the measured N2O with the simulated turbulence. We reformulated the argumentation from To conclude, the model is able to represent turbulence at a reasonable position (and time)

To conclude, the model grid scale wind field is consistent with the model turbulence scheme and can detect the occurrence of turbulence in the model.

L29 local changes in the energy budget

Corrected.

L30 affecting-; affect?

Corrected.

L67 the MESSy-fied - MESSy abbreviation has not been defined yet

Corrected

L129 conducted in the supplement

Corrected

L131 ..hence the mixing - a verb is missing?

Corrected to hence the mixing is more frequent

.. the Ellrod index does not fully representing?

Corrected to fully represent

Figure 5 and a majority of the following figures - the units are missing! Also what is the meaning of the (kg-1kg-1) unit? If the unit is indeed correct, why not to write kg-2?

The units is now added and corrected.

Caption of Fig. 5:.. difference (vdiff on - off) (c) Inverted O3-like tracers, (d)O3-like tracers at 2016-02-05 18:00. - Please rephrase for clarity.

Changed to and (c) difference (vdiff on - off) of the Inverted O3-like tracers (mol/mol), (d) difference of the O3-like tracers (mol/mol) at 2016-02-05 18:00.

L180-181:.. as the cross section of Figure 5. - in Figure 5? Btw. where is the cross-section located?

It is indicated in the map of geopotential height at the section of synoptic situation. It is now merged into each case studies for clarity.

L187 Figures 6c and 6d show Sections 3.3.2-3.3.4. define the tracer differences and the whole delta tracer-tracer correlation by equation and discuss the results more rigorously.

Figure 6c and 6d are the tracer-tracer correlation instead of delta tracer tracer correlation.

The intention of these graphs are to guide the readers before directly diving into the new delta tracer-tracer correlation. And therefore it is only discussed briefly.

L205.. The delta tracer-tracer correlation can also be used to determine the direction of vertical mixing. - but later in the text you mention only 1) no vertical mixing and 2) bi-directional mixing scenarios! Please pay attention to proper definition and discussion of mixing regimes, including balanced/unbalanced and reversible/irreversible regimes invoked later in the text. The part of the manuscript showing the results of delta tracer-tracer decomposed according to dynamical quantities and then normalized by the gradient is the strongest part of the paper that can elucidate how the turbulent parameterization works. But, more detail is needed how the plots were constructed.

They are now explained in the new section delta tracer-tracer correlation. We added in the manuscript In order to investigate the tracer mixing in the UTLS, we introduced a novel diagnostic, namely a delta tracer-tracer correlation, which is a similar concept to the tracer-tracer correlation, but makes use of the model capabilities. While the tracer-tracer correlation can be compared to the real world, the delta tracer-tracer correlation is a correlation between the differences of the tracers from model experiments. Instead of showing the mixing as an accumulation affected by other processes, it shows the impact of a single process (and potential subsequent advection differences). It requires 2 pairs of tracers (one pair of stratospheric and one pair of tropospheric). The difference of each pair, is a particular process being deactivated on one of the tracers in the model to investigate the impact of it. In our study, it is the turbulent vertical diffusion (vdiff). The detailed released tracers are described in section 3.2. The delta tracer-tracer correlation can also be used to determine the direction of vertical mixing. Several distributions are expected for different scenarios: (1) Concentrated distribution at the center [0,0] if no vertical mixing takes place at all; (2) Diagonal distribution for bi-directional mixing, where both tracers change at a similar rate, causing the data point spread along the diagonal. The bi-directional mixing could be either balanced or imbalanced, meaning an even (case 1, spread equally from the center [0,0]) or uneven (case 2, spread unequally from the center [0,0]) spread along the diagonal. Balanced bi-directional mixing indicates a similar amount of stratospheric tracers being exchanged with the tropospheric tracers, while imbalanced bidirectional mixing indicates a different amount of stratospheric tracers being exchanged with the tropospheric tracers. It could be attributed to different situations, details are discussed in the following cases. The upper left section of the diagram indicates the downward mixing of stratospheric tracers into the troposphere since at the same grid, there are increasing stratospheric tracers and decreasing tropospheric tracers. And the lower right quadrant indicates the opposite, with decreasing stratospheric tracers and increasing tropospheric tracers i.e. upward mixing of the tropospheric tracers. Scatter further away from the center indicates irreversible mixing, as the composition of the air masses is substantially modified, and the tracer is mixed irreversibly into the grid, i.e., instantaneously horizontally mixed. Additionally, this scatter is caused by initial differences from the mixing which are then amplified by (mostly horizontal) advection into regions where the vertical gradient of the tracers are different. Those different gradients can originate both from the tracer mixing event itself further upstream or from specific meteorological conditions, e.g., tropopause folds with strong gradients. Scatter away from the diagonal (case 3) indicates that the mixing occurs in a region with a different tracer gradient, a non-local effect introduced by other processes like horizontal advection acted on the mixed tracer. The scatter away from the diagonal gives an indicator that the mixing is non-local but the strength of mixing itself is still solely contributed by the turbulent mixing.

L243 ...figure 9d....figure 7d of case 1 have a similar N2 on both ends - You mean

**11d and 9d?**

Corrected

L255 2-¿two

Corrected

L256 s most likely due to the advection, considering the completely different wind field in Figure 8 and tropopause in Figure S17 - This feature has to be disentangled (maybe a 3D resolved overturn encompasses the unresolved turbulence region) and the methodology suggested by Ref3 can help here.

We further compared the TKE and the mixing (difference of the tracer with/without vertical diffusion), the mixing is located at the downwind region of the high TKE values region, unlike the other two cases, where the mixing has a better correlation with the model TKE. This also proves what we replied to reviewer 3 that the mixing fluxes themselves do not necessarily represent the mixing since they only show the instant mixing effect but ignore the post-mixing accumulation by the meteorology. We now reformulate in the manuscript from

The scatter away from the diagonal unlike the other two cases is most likely due to the advection, considering the completely different wind field in Figure 8 and tropopause in Figure S17, the strong horizontal advection in the region of strong horizontal gradients changes the background ratios in addition to the vertical mixing and thus introduces additional mixing during each time step compared to the other cases.

to

The scatter away from the diagonal unlike the other two cases, where the modeled TKE is better correlated with the mixing (not shown), is due to the advection, the mixing shown in Figure S17 and S22 located at the downwind region of the high TKE region (Figure S21). In the earlier time, the mixing region (Figure S22, left panel) is more co-located with the high TKE region (Figure S21, left panel). After several hours, the mixing region (Figure S22, right panel) propagates to the downwind region while the high TKE region (Figure S21, right panel) remains at the same location. The strong horizontal advection in the region of strong horizontal gradients changes the background ratios in addition to the vertical mixing and thus introduces additional mixing during each time step compared to the other cases. The wider the scatter is, the more, e.g., tropospheric tracer depletion is found at similar stratospheric tracer values.

**L262 ... such that the mixing is almost equally balanced. What does this statement about the mixing type mean?**

The mixing type is now mentioned in the new section delta tracer-tracer correlation in the manuscript Balanced bi-directional mixing indicates a similar amount of stratospheric tracers being exchanged with the tropospheric tracers, while imbalanced bi-directional mixing indicates a different amount of stratospheric tracers being exchanged with the tropospheric tracers.

L272-273 To conclude, vertical turbulent mixing by CAT in the model simulations leads to an enhanced and significant tracer mixing in the UTLS region. - Is there any novelty in this statement? This seems to come from the definition of the turbulence parameterization.

Even though the general statement is not new, we now have a systematic proof that the model has a suitable representation of turbulence in the UTLS and it has a corresponding effect on tracer redistribution.

L275-276 Strong dynamical forcing like vertical wind shear could lead to mixing even in the stable atmosphere with a typical stratospheric N2 value. - This is a textbook knowledge, intensively studied under the term dynamic instability?

We now changed in the manuscript to This confirms the findings of Kaluza et. al. (2021) and Kunkel et al. (2019) that strong dynamical forcing like vertical wind shear could lead to mixing even in the stable atmosphere with a typical stratospheric  $N^2$  value.

**L280 provides a reliable tool - this was not assessed in the manuscript**

We think that the model simulation now shows that we have reasonable turbulence and associated mixing, considering we have now added comparisons with the observation, we think it is a suitable tool. The manuscript now changed to provides a suitable tool.

L283 ..MECO(n) is presented.-; MECO(n), is presented.

Corrected

L286.. balanced and imbalanced bi-direction mixing - balanced/imbalanced mixing has not been defined/introduced before. I had the impression that the motivation was to analyze the direction of mixing. Nevertheless, there is a great potential to define the different mixing regimes robustly and provide a systematic analysis on these aspects in the revision

We have added in the manuscript Balanced bi-directional mixing indicates a similar amount of stratospheric tracers being exchanged with the tropospheric tracers, while imbalanced bi-directional mixing indicates a different amount of stratospheric tracers being exchanged with the tropospheric tracers. in the section introducing the delta tracer-tracer correlation.

L287-289 The simulated turbulent kinetic energy (TKE) is spatially and temporally well matched with the (post-simulation) diagnosed Ellrod Index, showing the model is able to generate turbulence in the UTLS in agreement with the gridscale wind field data from the model output. - ; I assume that the Doms et al. parameterization has been validated before against observations.

We do not find any proper documentation where the UTLS turbulence in COSMO was compared against observation, since it was mostly designed and tested for the boundary layer. And sparse studies are focusing on the performance of the turbulence scheme in UTLS. We can only find one study from Muñoz-Esparza et. al. (2020) that tested it for WRF.

L293-294 ...which located near the tropopause (case 1) experiencing the strongest mixing considering the high vertical wind shear and tracer gradient. -; please rephrase.

Changed to which located near the tropopause experiencing the strongest mixing due to the high vertical wind shear and tracer gradient (case 1)

L295.. when measurement data is available. - you mean that there are not any data even from targeted campaigns existing that would be suitable for validating

**the model?**

Some measurement is available, but an ideal measurement data would be something like a pseudo-lagrangian measurement which can better distinguish between the mixing and the post-mixing effect. We now changed to when a more comprehensive measurement dataset is available.

L300 .. both, the forcing or the pre-existing tracer gradients are the dominant drivers for the exchange. -¿please rephrase

Change from Depending on the individual situation, both, the forcing or the pre-existing tracer gradients are the dominant drivers for the exchange.

to

Depending on the individual situation, either the dynamical forcing or pre-existing tracer gradients (or both) can be the dominant drivers for the exchange events.

L302-305 These events can be irreversible, i.e., the exchange of tracers happens along the diagonal of a delta tracer-tracer correlation, leading to a disturbance of typical stratospheric or tropospheric chemical compositions in the respective parts of the atmosphere with implications for climate, e.g., via the radiative impact of exchanged species. - From the definition of a delta tracer-tracer correlation, these events are unresolved in the model and mediated by the parameterization, hence inherently irreversible. However, is this unresolved process the only irreversible mixing process in the model given that your resolution may be sufficient for resolving parts of the overturning?

In MECO(n), other sub-grid scale processes like gravity waves could potentially affect the dynamics of the turbulence, but the gravity waves themselves do not affect the tracer mixing ratios directly. As we are switchting off the turbulence mixing for the specific tracers, the turbulence is the only irreversible mixing process for tracers in the model, beyond the subsequent advection which amplifies the changes originating from the parameterised vertical diffusion.